# The OMZ and nutrients features as a signature of inter-annual and low-frequency variability off the Peruvian upwelling system

Michelle I. Graco [1], Sara Purca [1], Boris Dewitte [2,3,4], Carmen G. Castro[5], Octavio Morón[1], Jesús Ledesma[1], Georgina Flores[1], Dimitri Gutiérrez[1]

[1]Dirección General de Investigaciones Oceanográficas y cambio Climático. Instituto del Mar del Perú (IMARPE), P.O. Box 22, Callao, Perú.
[2]Laboratoire d`Etudes en Géophysique et Océanographie Spatiale (LEGOS)/IRD, Toulouse, France
[3]Universidad Católica del Norte, Facultad de Ciencias del Mar, Coquimbo, Chile
[4]Centro de Estudios Avanzado en Zonas Áridas (CEAZA), Coquimbo, Chile
[5]CSIC, Instituto de Investigaciones Marinas, Eduardo Cabello 6, 36208 Vigo, Spain.

*Correspondence to*: Michelle I. Graco (mgraco@imarpe.gob.pe)

**Abstract.** Over the last decades, the Humboldt Current Upwelling Ecosystem and particularly the Northern component off Peru, has drawn the interest of the scientific community because of its unique characteristics: it is the upwelling system with the biggest catch productivity, despite the fact it is embedded in a shallow and e intense Oxygen Minimum Zone (OMZ). It is also an area of intense nitrogen loss and anammox activity and experiences a large inter-annual variability associated with the equatorial remote forcing. In this context, we examined the oceanographic and biogeochemical variability associated with the OMZ off central Peru, from a monthly time series (1996–2011) recorded off Callao (12º02'S, 20 nm). The data reveals a rich spectrum of variability of the OMZ that includes frequencies ranging from seasonal to inter-annual scales. Due to the efficient oceanic teleconnection off Peru, the observed variability is interpreted in the light of an estimate of the equatorial Kelvin wave contribution to sea level anomalies considering peculiarities of its vertical structure (i.e. first two baroclinic modes). The span of the data set allows contrasting two OMZ regimes. The "strong" regime associated with the strong 1997–1998 Equatorial Pacific El Niño, during which the OMZ adjusted to Kelvin wave-induced downwelling conditions that switched off the upwelling and drastically reduced nutrients availability. The "weak" regime corresponding to the post-2000 period associated to the occurrence of moderate Central Pacific El Niño events and enhanced equatorial Kelvin wave activity, in which mean upwelling conditions are maintained. It is shown that the characteristics of the coupling between physics and biogeochemistry is distinct between the two regimes, with the "weak" regime being associated to a larger explained variance in biogeochemical properties not related to the ENSO oceanic teleconnection. The data also reveals a long-term trend from 1999 corresponding to a deepening of the oxygen deficient waters and a warming. Implications of our results for understanding the OMZ dynamics off Peru are discussed.

## 1 Introduction

The upwelling region off Peru hosts a complex biogeochemical system that is unique for at least two reasons. First, it is embedded into the permanent, intense and shallow Oxygen Minimum Zone (OMZ) of the Eastern Tropical South Pacific (Gutiérrez et al., 2008). Second, it exhibits a significant variability at different time scales, particularly at inter-annual scale associated with the Equatorial Kelvin waves and the El Niño-Southern Oscillation, ENSO (Chavez et al., 2008).

The OMZ is generated by the combination of high oxygen demand during organic matter remineralization and the sluggish ventilation in the region (Wyrtki, 1962; Helly and Levin, 2004). It is wide in the vertical extension (~ 500 m), intense (< 22.5 $\mu$mol kg$^{-1}$), and at some latitudes the upper boundary could be very shallow (25-50 m) intersecting the euphotic zone and

impinging the continental shelf (Morales et al., 1999; Schneider et al., 2006; Fuenzalida et al., 2009; Paulmier and Ruiz, 2009;
Ulloa and Pantoja, 2009). The OMZ off Peru is associated with the presence of nutrient-rich Equatorial Subsurface Water
(ESSW) transported poleward by the Peru-Chile Undercurrent (PCU) (Strub et al., 1998; Fuenzalida et al., 2009; Silva et al.,
2009) and the transport of low-oxygenated waters by the narrow primary and secondary Southern Subsurface Countercurrents
near 4°S and 7°S respectively also known as Tsuchida jets (Furue et al., 2007; Montes et al., 2010). Recent modeling studies
also highlight the important role of sub to mesoscale dynamics in constraining the upper OMZ meridional boundaries
(Bettencourt et al., 2015; Vergara et al., 2016).

The OMZ variability in terms of distribution and intensity has a direct impact on the biogeochemical processes of the

northern region of the Humboldt upwelling system, because oxygen: 1) is a key factor in biogeochemical cycles, particularly
in the carbon (Friederich et al., 2008) and nitrogen processes (e.g. Kock et al., 2016; Hammersley et al., 2007; Lam and
Kuypers, 2011; Dale et al., 2017), 2) its consumption determines high nitrogen loss and in consequence low N:P ratio of
upwelled waters, below the classical Redfield ratio of 16, with a strong impact on the primary and secondary production (Franz
et al., 2012) and 3) is a control factor in the distribution of organisms (e.g. Bertrand et al., 2010; Criales-Hernández et al.,
2006; Ekau et al., 2010; Gutiérrez et al., 2008, Levin et al, 2002). The position, strength, and thickness of the Eastern South
Pacific OMZ can be greatly modified by local and/or remote forcing (e.g. inter-annual time scales, Morales et al., 1999;
Gutiérrez et al., 2008). During ENSO episodes, equatorial fluctuations in sea level and currents propagate along the Peruvian
coast, which behaves as an extension of the equatorial waves guide (Clarke and van Gorder, 1994). Strong El Niño (EN)
events, like the 1997-1998 Eastern Pacific El Niño, affect the circulation and water masses distribution causing the deepening
of the OMZ and the occurrence of large oxygenation events in the water column and over the sediments along the Chilean and
Peruvian coast (Morales et al., 1999; Sánchez et al., 1999; Gutiérrez et al., 2008). In fact, Helly and Levin (2004) reported
that, during the 1997-1998 El Niño, about 61% of the OMZ volume off Peru and northern Chile was reduced.

While most studies on the inter-annual impact have focused on the 1997-1998 El Niño, recent studies indicate that

the characteristics of the inter-annual variability have changed in the last decades. In fact, since the 90s a higher frequency of
the so-called Central Pacific (or Modoki) El Niño events occurs (Yeh et al., 2009; Lee and McPhaden, 2010; Takahashi et al.,
2011). This type of El Niño event does not undergo a significant warming of Sea Surface Temperature (SST) along the coast
of Peru conversely to Eastern Pacific El Niño events (Ashok et al., 2007; Dewitte et al, 2012). On the other hand, Central
Pacific El Niño events are associated with strong activity of Intra-seasonal Equatorial Kelvin Waves (IEKW) (Mosquera et al.,
2014) that can lead to thermocline depth fluctuations along the coast of Peru through the propagation of coastally trapped
Kelvin waves (cTKW) (Clarke, 1983; Dewitte et al., 2011; Illig et al., 2014).

The study of the relationship between ENSO and the OMZ therefore, would require taking into account the different

time scales of variability along the equator, from intra-seasonal to inter-annual. Most existing studies have documented the
physical properties (Morón O., 1991; 2000) and chemical properties of the waters along the coast of Peru in relation with the
inter-annual equatorial variability (Calienes and Guillén, 1981; Guillén and Izaguirre de Rondán, 1973; Guillén et al., 1989;
Ledesma et al., 2011) disregarding the higher-frequency time scales and the diversity of ENSO (Capotondi et al., 2015). Here,
we analyze a unique long-term time series of oxygen and inorganic nutrient data off central Peru, Callao, spanning 16 years
(1996-2011). The region of Callao has been identified as one of the major upwelling cells off central Peru (Rojas de Mendiola,
1981) with a well-developed OMZ at subsurface (Wooster and Gilmartin, 1961; Zuta and Guillén, 1970). The presence of
nitrate-rich ESSW (Zuta and Guillén, 1970; Strub et al., 1998; Graco et al., 2007; Silva et al., 2009) triggers the high primary
production of the region, with maximum values in spring-summer, out of phase of winter upwelling maximum (Echevin et al.,
2008; Chavez and Messié, 2009; Gutiérrez et al., 2011, Pennington et al., 2006; Vergara et al., 2016).

This dataset offers the opportunity to get insights in the ENSO oceanic teleconnection on the OMZ and nutrients

features off the Peruvian upwelling system, considering recent advances in our understanding of ENSO events (Capotondi et
al., 2015). The period under consideration encompasses characteristic events of the two ENSO regimes described by Takahashi
et al. (2011), that is a strong Eastern Pacific El Niño (i.e. the 1997-1998 strong El Niño) and a series of moderate Central
Pacific El Niño events after 2000. Finally, the study explore the long-term trend from 1999 corresponding to a deepening of
the oxygen deficient waters and a warming. Implications of our results for understanding the OMZ dynamics off Peru are
discussed.
**2 Material and Methods**
**2.1 Study site and data**

The study site corresponds to a station off Callao (central Peru 12°02' S, 77°29´ W, Fig. 1) located 20 nm from the coast and
with 145 m water column depth. The station was visited most of the time by the Instituto del Mar del Peru (IMARPE) ship on
a monthly or bimonthly basis between 1996 and 2011 to carry out vertical profiles of temperature, salinity, oxygen and nutrients
(nitrate, nitrite, phosphate and silicate). Gaps larger than one or two months are however present over this period with the year
2011 having 30% of missing data, which results in some limitations and require caution in processing the data and in
interpreting the results (see section 2.2).
The temperature was measured by inversion thermometer through 2001 and by CTD (Seabird SBE 19+) from 2002.
Salinity was measured by salinometer through 2001 and by CTD plus salinometer from 2002. Comparison between CTD
measurements and estimates of temperature and salinity derived from water samples from the Niskin bottles were made
regularly during all the cruises to verify the proper calibration of the instruments.
Dissolved oxygen and nutrients were measured most of the time at standard depths (0, 10, 30, 50, 75, 100 m). Dissolved
oxygen was determined by a modified Winkler method (Grasshoff et al., 1999), with a precision of 0.5 µmol kg$^{-1}$. Nutrient
samples (nitrate, nitrite, phosphate and silicate) were frozen and stored before being analyzed using standard colorimetric
techniques (Parsons et al., 1984). The estimated accuracy of the method was $\pm$0.5 µmol L$^{-1}$ for nitrate, $\pm$0.08 µmol L$^{-1}$ for
nitrite, $\pm$ 0.03 µmol L$^{-1}$ for phosphate and $\pm$0.25 µmol L$^{-1}$ for silicate.Fixed nitrogen deficit (Ndef) was determined by the
formula:
$Ndef= 12.6 \times [ HPO_4^{2-} ] – ([NO_3^-]+[NO_2^-])$
The constant 12.6 is the empirically-determined N:P ratio of organic matter produced in these waters (Codispoti and
Packard, 1980). Positive values indicate nitrate deficit.
The OMZ was defined as the area with oxygen concentrations lower than 22.5 µmol kg$^{-1}$. This concentration was considered
as the OMZ upper boundary (Schneider et al., 2006; Fuenzalida et al., 2009; Ulloa and Pantoja, 2009).
**2.2 Statistical analysis of Time series**

The physical and chemical data off Callao showed missing monthly data, in particular the 2011 year that present up to 30% of
the data missing. This is an inherent limitation of our data set that we have to take into account for the interpretation of the
variability. In particular, the intra-seasonal variability (periods ranging from one month to 3-4 months) associated to the
aliasing induced by the sampling of the data (i.e. one data point at best per month). Since the environmental conditions in the
study region vary at daily to intra-seasonal time scales (Dewitte et al., 2011), the approximation that one measurement yields
a monthly mean data may be biassed. For this reason, the intra-seasonal variability in the data will not be documented in the
paper that will focus on inter-annual time scales. As a consistency check and as an attempt to overcome such a limitation, we
will use two methods to fill the gaps. First, the data were processed by linear interpolation in the vertical at each times step
where data are available, then data either linearly interpolated in the time domain (first method) or using a 6-month running
mean filter (second method). At least 2 data points within the 6-month windows are required, which leads to a data set without
gaps between January 1996 (data over 1995 are used to be able to start in January 1996 using such a filtering) and September
2010. The latter procedure results in a low-pass filtering of the data so that aliasing and gridding artifacts are reduced compared
to the first method. The first method is used to derive the oxycline and thermocline and second method is used to derive
anomalies relative to a mean climatology. The latter is calculated from the raw data that have been only interpolated vertically
at each time step on a regular grid (levels are 0, 10, 25, 50, 75 and 100 m). The period for the calculation of the climatology is
1999-2011 (i.e. 13 years) but, due to the gaps in the time domain, each calendar month is calculated over a different number
of years, always larger than 4 and lower than 12. The anomalies were calculated as the difference between the low-pass filtered
data and the mean climatology. Wavelet analysis are performed on the time series and the global wavelet spectrum is derived
(Torrence and Compo, 1998).

The Empirical Orthogonal Function (EOF) analysis (Thomson and Emery, 2014) was applied to the combined

normalized time series of temperature, salinity, oxygen, inorganic nutrients, to extract the statistically dominant mode of
covariability between the different components of OMZ dynamics (i.e. physical versus biogeochemical). The normalization
of the time series consists in dividing them by their standard deviation. The analysis was performed taking into account the
time series at all depths between 5m and 100m so that the statistics grasp some aspects of the vertical structure variability. The
Pearson correlation coefficient (r) was calculated between the data (or their PC time series) and some indices (see section 2.3)
and the significance level of the correlation was estimated based on the degree of freedom inferred from the autocorrelation of
the time series (i.e. taking the lag when it reaches zero the first time).

Long-term linear trend was calculated from the data that have been only interpolated vertically at each time step on

the regular grid so that gaps in the time domain are considered in the estimate of the trend. They were calculated from January
1999 to avoid an artifact associated with the strong 1997-1998 El Niño event. We tested whether the value of the slope is
significantly different from 0 based on a Student's t-test. If the significance level is lower than 80%, the trend is considered
not significant and only the slope values with confidence level larger than 90% are discussed.

**2.3 El Niño indices and Equatorial Kelvin Waves (EKW)**

In order to select the El Niño and La Niña years, we use the Oceanic El Niño Index (ONI) provided by the national Weather
Service NOAA (NOAA, CPC. 2015). The magnitude of the events (weak, moderate, strong) are based on the ONI index, while
the type of events (Eastern Pacific (EP) versus Central Pacific (CP) El Niño event) follows Yu and Kim (2013). This
information is summarized in Table 1. To investigate the relationship between ENSO and the variability in the data, we use
two other indices to account for the large-scale inter-annual variability in the equatorial Pacific. The two indices, named E and
C, were defined by Takahashi et al. (2011), and account respectively for the Eastern Pacific El Niño events (hereafter EP
events) and the Central Pacific El Niño events (hereafter CP events). These two indices are obtained from the first two PC
time series of the EOF modes of the SST anomalies in the tropical Pacific (HadISST data set, Rayner et al.(2003)) over the
period 1950-2014. Over the period of interest (1996-2011) the two indices are uncorrelated, so that they can be conveniently
used as a basis over which the variability in the data can be projected. The projection onto the E index will account for the
share of the variability in a particular field (T, S, $O_2$, Nitrate or Nitrite) that is associated to EP events, while the projection
onto the C index will quantify the relationship between CP events and the variability in the data. An approximate temperature
(similarly for S, $O_2$, Nitrate and Nitrite) field can therefore be derived through bilinear regression analysis: $T_{approx}(z,t)=$
$<T(z,t)|E(t)>.E(t)+<T(z,t)|C(t)>.C(t)$. It is expected that if there is a strong linear relationship with ENSO, the approximate
data will explain a significant variance of the original data. However this may be not always the case for two reasons: 1) The
relationship between ENSO and the variability in the data is not necessarily linear. 2) The coastal data are influenced by time
scales of variability that are not necessarily accounted for by the E and C indices, since the latter are derived from SST which
is less variable than thermocline fluctuations (cf. Dewitte et al. (2008) for the equatorial region). In particular, Kelvin wave
activity is enhanced several months prior to the ENSO peak (Mosquera et al., 2014) and the former can have a strong impact
on the coastal circulation (cf. Ramos et al. (2008) for the 1997/98 El Niño event) although the ENSO peak is not already
reached. That is why we will also use an estimate of the Equatorial Kelvin wave (EKW) activity to account for aspects of the
remote forcing not necessarily contained in the E and C indices.
The estimate of the EKW was derived from an Ocean General Circulation Model (OGCM) simulation provided by
Mercator, the European Institute for operational oceanography. The simulation has been validated from observations in recent
previous studies (Mosquera-Vásquez et al., 2014), which indicates that, despite not assimilating observations, the simulation
is as realistic as these SODA oceanic reanalysis (Carton and Giese, 2008) in the near-equatorial region over their overlapping
period (1992-2008). The use of an OGCM simulation to derive the equatorial Kelvin wave is motivated by the possibility to
estimate their vertical structure and separate waves having different propagating characteristics (phase speed and amplitude),
which is not possible from observations. The method for deriving the wave coefficient consists in projecting the pressure and
current anomalies from the model between 15°S and 15°N onto the theoretical vertical mode functions obtained from the
vertical mode decomposition of the mean stratification. Kelvin wave amplitude is then obtained by projecting the results onto
the horizontal modes at each grid point in longitude. The method has been shown to be successful in separating first and second
baroclinic waves (Dewitte et al., 1999, 2008) that propagate at different phase speeds and impact the Peru coast in a very
specific way (Illig et al., 2014). In particular, due to the sloping thermocline from west to east along the equator, the second
baroclinic mode Kelvin wave tend to be more energetic and influential on the upwelling variability off the Peruvian coast
(Dewitte et al., 2011, 2012). For the correlation analysis with the dissolved oxygen data, we select the EKW amplitude (in cm)
for the first and second baroclinic modes (hereafter EKW_1 and EKW_2) at 90°W. Considering the phase speed of a coastal
trapped-Kelvin wave (~250 km/days) this can be assumed as a good proxy of what happens off Callao in terms of wave activity.

**3 Results**

**3.1 Temperature and Salinity**

The evolution of temperature and salinity off Callao over 1996-2011 as a function of depth is shown in Figure 2 a and b.
Temperature and salinity experience significant temporal fluctuations at a wide range of time scales. Fluctuations,
corresponding to near annual and inter-annual periods, are observed in the thermocline depth (15° C isotherm). Several
deepening of the thermocline take place that correspond most of the time (but not always) to El Niño episodes and that are
associated to enhanced salinity. There is in particular a strong deepening (more than 100m) of the 15°C isotherm between 1997
and 1998 associated with significantly saltier waters over the water column (>35.1), which correspond to the signature of the
strong 1997-1998 El Niño event. Note also that the disappearance of the 15°C isotherm in the upper 100 m that took place in
early 1997 (around April), well ahead of the El Niño peak phase (around November). A slight deepening of the thermocline
takes place also at the beginning of 2002, 2005, 2010 and during winter of 2006, 2008, 2009 and 2011. This thermocline
deepening was coincident with high salinity values, clearly detected over the period 2002-2007.
The temperature data are further decomposed into a mean seasonal cycle and anomalies relative to this seasonal cycle
(Figures 3 a and b). The latter is calculated over the period 1999-2011 in order to avoid a bias in the statistics due to the
presence of the strong 1997-1998 El Niño. The mean seasonal cycle indicates that during austral winter and spring (June-
August) colder conditions prevail in the water column, which is associated with less saline conditions (34.8-35.1). In particular,
the 17°C isotherm outcrops between July and November. Note that during winter a slight deepening of the thermocline takes
place as evidence by the evolution of the 15° C isotherm, while during spring the thermocline is shallower (up to 20 m). The
climatology also reveals the existence of a semi-annual component, which is thought to be related to the semi-annual cycle of
the equatorial variability through oceanic teleconnection (Ramos et al., 2006).

The temperature and oxygen anomalies (Figure 3a) highlights the rich spectrum of variability as well as the
differences between the strong 1997-1998 El Niño, a EP event and the subsequent periods. While temperature anomalies reach
~5°C at all depths during the 1997-1998 EP El Niño event, they are much weaker (~1°C) for the following events and do not
extend as deep. The period after 2000 corresponds to a period of dominant CP El Niño events that have a weaker amplitude
than the 1997-1998 EP El Niño (Table 1), which explains the weaker temperature anomalies off Callao. A noticeable feature
of the temperature anomalies after 1999 is also the existence of a long-term warming trend with a deepening of the 15°C
isotherm estimated to -0.30 m decade $^{-1}$. The analysis of the seasonality of the trend indicates that there is long-term deepening
of the thermocline summer, spring and fall, while this is not the case in winter. The analysis of the trends (1999-2011) as a
function of depth indicate that the warming takes place over the whole water column except at the surface (see Table 2).
**3.2 Dissolved oxygen**

The time series of dissolved oxygen over 1996-2011 off Callao exhibits a similar evolution than the thermohaline time series
previously shown (Figure 2c) with large fluctuations at inter-annual scales. The strong 1997-1998 El Niño event is associated
with the largest and deepest oxygenation event over the entire record (> 100 μmol kg$^{-1}$). A slight oxygenation in the water
column takes place during 2002, 2006, 2008 2009 and 2011. This oxygenation is evidenced in the position of the oxycline
(iso-oxygen of 45 μmol kg$^{-1}$) and OMZ upper boundary (22.5 μmol kg$^{-1}$ iso-oxygen) depth.
The climatology of dissolved oxygen (Figure 3d) indicates a seasonal and semi-annual component, since the oxycline and
OMZ upper boundary depth is shallower in summer- early fall (20-40 m) and spring (up to 20 m) and deeper in winter (50 m).
Note that during summer-early fall and spring, oxygen-poor waters can intercept the euphotic layer and also the continental
shelf, promoting suboxic and even anoxic conditions in bottom water underlying surface sediments (O$_2$ < 8.9 μmol kg$^{-1}$; Fig
2c).

The time series of dissolved oxygen anomalies (Fig 3c) presents large positive oxygen anomalies during El Niño
events, as the intense oxygen anomaly (> 60 μmol kg$^{-1}$) associated with the strong 1997-1998 El Niño when well-oxygenated
Subtropical Surface Waters (SSW) occupied the upper 100 meters (Morón et al., 2000). The close relationship between the
oxygen and thermocline depth during the 1997-1998 El Niño (i.e. positive O$_2$ anomalies associated to a deepening of the
thermocline) breaks down afterward for some events. Before 2000, the OMZ depth present a significant correlation with the
15ºC isotherm depth (r= 0.61, v-p < 0.01). After 2000, the correlation drops down (r = 0.28, v-p < 0.01).

The position of the OMZ upper limit shows a negative trend after 1999 with a deepening estimated to -0.64 m decade
$^{-1}$ (see Table 2) that points to a long-term deepening of the oxygen deficient waters, similar to the deepening trend of the
thermocline depth for the same period. No significant difference appears in the seasonality of the OMZ trend (Table 2), contrary
that the thermocline (15° C depth) trend. The long-term trend (1999-2011) of oxygen concentration at different depths indicates
a significant increase of oxygen in the entire water column. At surface, the increase is 24.03 μmol kg$^{-1}$ decade $^{-1}$ (Table 2) while
maximum trend is found in the first 25 m (47.55 μmol kg$^{-1}$ decade $^{-1}$).

**3.3 Nutrients and biogeochemical activity**

The time series of inorganic nutrient vertical distributions off Callao are shown in Figure 4. Nitrate and nitrite concentrations
ranged from ca. 0.0 to 27.0 μmol L$^{-1}$ and ca. 0.2 to 9.0 μmol L$^{-1}$ respectively. Lower nitrate values are present at the surface (5
m depth) and at deep waters (> 80 m depth), particularly during summer and fall periods, while maximum nitrite values appear
at subsurface waters in opposite relationship with nitrate levels. During winter, high nitrate concentrations are found in the
entire water column (> 15 μmol L$^{-1}$). The vertical distributions of silicate and phosphate exhibit a similar pattern than nitrate
(not shown).
Nutrient data also present a strong inter-annual signal particularly during the strong 1997-1998 El Niño event with
low nitrate concentrations (< 10 μmol L$^{-1}$) coincident with minimum and even zero nitrite values and low silicate and phosphate
levels (<10 μmol L$^{-1}$ and 1 μmol L$^{-1}$ respectively; Fig. 4). Between 1999 and 2001 nitrate concentrations were also lower than
10 μmol L$^{-1}$ on average, but in contrast with the previous El Niño episodes, subsurface nitrite reached maximum values (up to
9 μmol L$^{-1}$) coincident with an intense OMZ development and shallow thermocline (Fig. 2c).
Low silicate levels are registered at the sea surface, while silicate concentrations > 25 μmol L$^{-1}$ associated with
phosphate concentrations > 3 μmol L$^{-1}$ are registered in subsurface waters. After 2000, subsurface nitrate concentrations
reached the highest values (> 20 μmol L$^{-1}$) over the entire record that is coincident with lower nitrite and silicate concentrations
(Fig. 4). High nitrite pools in the water column were described as a typical feature under oxygen deficient waters (Deuser et
al., 1978) and a tracer of denitrification (Codispoti and Packard, 1980; Codispoti and Christensen, 1985; Codispoti et al. 1986;
1988) and anammox activity (Hammersley et al., 2007; Lam et al., 2009; Lam and Kuypers, 2011) off Peru.
In order to explore some biogeochemical activity related to the nitrogen cycle and the OMZ variability off Callao,
Ndef in the water column is estimated (Fig. 5). Ndef values range from negative (-5 μmol L$^{-1}$), indicative of low nitrate
consumption, up to 40 μmol L$^{-1}$ corresponding to conditions of high nitrate deficiency. Ndef exhibits a clear inter-annual signal
with minimum values (zero-negative) during the 1997-1998 El Niño event coincident with well-oxygenated waters (Fig. 2c).
Low values of Ndef associated with almost zero nitrite concentrations suggest lower denitrification and/or anammox activity
during these strong El Niño events. The significant effect on denitrification in the eastern South Pacific Ocean due to changes
in the equatorial winds during El Niño was previously described by Codispoti et al. (1988). On the continental shelf off Callao,
Dale et al. (2017), showed the occurrence of intra-annual and inter-annual variability in the denitrification and anammox
processes that appears to decrease under El Niño coupled with low primary productivity and high bottom waters oxygenation.
In contrast, between 1999 and 2001, Ndef of 40 μmol L$^{-1}$ peaked under a shallow and well-developed OMZ (Fig. 2c), pointing
to nitrate reduction as an important route of organic matter remineralization yielding high nitrate concentrations at subsurface
waters (Fig. 4).
After 2000, Ndef water column conditions were highly variable coincident with the variability in the OMZ
distribution and in general a less intense OMZ. Intense nitrate deficient conditions were registered in 2005, 2007 and at the
end of 2011 coincident with La Niña conditions (Table 1). Ndef at subsurface (50 and 90 m depth) was significantly correlated
with the 15ºC isotherm depth (r= 0.43, v-p < 0.01) and with the OMZ though the correlation is relatively low (r= 0.28, v-p <
0.01). The Nitrate and Nitrite data do not exhibit a significant long-term trend from 1999 over the water column (See Table 2)
although the post-2000 period has characterized by a reduction (increase) of nitrite (nitrate).

**3.4 Equatorial forcing, Kelvin wave activity (EKW) and the OMZ off central Peru**

As a first step, we document here the linear relationship between the variability in the data and ENSO taking into accounts its
diversity (i.e. the existence of EP and CP El Niño events). This consists in regressing the data onto the E and C indices, which
yields an approximate data that corresponds to the result of a statistical model where the E and C indices are predictors (see
section 2). The regression coefficients for each data is presented in Figure 6 in dimensionalized unit, which provides the range
of variations of a given field during a strong EP El Niño event (red curves in Fig. 6) and during a CP El Niño event (blue
curves in Fig. 6). Note that the C index also accounts for the La Niña events so that the blue curves can be also interpreted as
the variations of a particular field during a cold event. The Figure 6 reveals that variations in temperature, salinity and oxygen
during EP El Niño events are larger (~twice) than during CP El Niño event and of similar sign, consistently with the above
description (Figures 2 and 3). The variation as a function of depth is also rather homogeneous, except for oxygen in the upper
layer. In contrast, the variations for nitrate tend to have opposite signs and the amplitude during EP El Niño events is weaker
in absolute value. The negative variation of nitrate during the EP El Niño events is consistent with the reduced upwelling
during such events which results the deepening of the reach-nutrients cold coastal waters. In contrast, during CP El Niño
events, nitrate concentration tends to increase consistently with the maintenance of upwelling conditions (see Dewitte et al.
(2012)). During both EP and CP El Niño events, nitrite tends to decrease with depth in a similar way that could be explained
by oxygen availability under both El Niño types.

The relevance of these results needs to be assessed in light of the variance explained by the statistical model, which

is provided in Figure 7. The latter indicate a significant amount of variance explained by the linear statistical model for
temperature and salinity (~30%). It is much less so for the biogeochemical fields, which could be due to two reasons: 1) there
is a significant contribution of non-linear processes in the coupling between physics and biogeochemistry at ENSO time scales
off Callao or/and, 2) the E and C indices do not account for all variability time Scales relevant for the oceanic teleconnection
off Peru onto the biogeochemistry (i.e. equatorial Kelvin wave). In order to get further insights in the remote forcing of the
OMZ, we thus document the evolution of EKW activity during 1996-2011. The evolution of the amplitude of the EKW_1 and
EKW_2 at 90°W in terms of sea level anomalies is shown in the Figures 8a and b. These waves transmit their energy along
the coast in the form of coastal trapped Kelvin waves and can trigger extra-tropical Rossby waves (Clarke and Shi, 1991). It
is assumed that waves with amplitude larger than one standard deviation over the study period are downwelling Kelvin waves,
whereas amplitudes more negative than -1 standard deviation correspond to upwelling Kelvin waves. The EKW_2 activity is
delayed by 1 month compared to EKW_1, consistent with the difference in phase speed of the waves and their propagation
from the central equatorial Pacific up to 90°W. Maximum correlation between both time series was before 2000 (r=0.67, v-p<
0.01), being significantly lower after 2000 (r=0.42, v-p< 0.01).

Positive anomalies of EKW are associated with a deepening of Z_15°C (<0), as was observed during the strong 1997-

1998 El Niño, the weak 2002- 2003 El Niño and during 2006, 2008 and 2010 warm seasons. During these periods, EKW_1
and EKW_2 are almost in phase with comparable amplitude and Z_15°C and Z_OMZ (Fig. 8 d) are out of phase. The EKW_1
and EKW_2 are highly correlated with Z_15°C and Z_OMZ variables, but we find that EKW_2 has a stronger relationship
with the Z_15°C and Z_OMZ (r -0.54, -0.40 respectively, v-p<0.01) than EKW_1 (r -0.34, -0.23 respectively, v-p<0.01).
The global wavelet spectrum of the EKW_1 and EKW_2 time series at 90°W show that the second baroclinic mode is
associated with lower frequency variability than the first baroclinic Kelvin wave mode (Fig. 8 a, b). It is noteworthy that
EKW_2 is negatively skewed since 2000 (normalized skewness = -0.8910 cm) and there is a negative trend of upwelling events
since 2000 (trend = -0.0177 cm/decade), features that are also encountered for Z_15°C (Fig. 8 c) (normalized skewness = -
1.330 m and trend = -0.30 m decade$^{-1}$). The global wavelet spectra for Z_15°C and Z_OMZ (Fig. 8 h, i) also reveal a rich
spectrum of variability that is characterized by a double peak, one at ~4 years and the other one near 2.2 years. N def at 50 m
(Fig. 8 d) exhibits a spectrum with a single dominant peak at ~4 years. Summarizing, the spectral analysis of the synthetic
proxies for the physical (EKW, Z15) and biogeochemical (Z_OMZ, Ndef at 50m) processes suggests that coupling between
physics and biogeochemistry in relation with ENSO may take place at various time scales.

As an attempt to quantify such a coupling and its relationship with the remote forcing, an EOF analysis is performed

combining all the data (temperature, salinity, oxygen, nitrate and nitrite) (see section 2 for details). The EOF analysis is
performed over two periods, 1996-2010 and 1999-2010 to differentiate the coupling characteristics as a function of the El Niño
types considering that the 1997-1998 EP El Niño event is very influential on the obtained statistics. The results of the EOF
analysis are presented in Figure 9 for the first and second EOF modes (EOF1 and EOF2). The PC time series for the period
1996-2010 indicate that the EOF1 mode mostly captures the variability (43%) associated to the strong 1997-1998 El Niño
event (with a high correlation between PC1 and the E index, see Table 3), while the EOF2 mode explanis 18% of the variance
and does not relate to any particular events or series of events. The latter is confirmed by the low correlation between PC2 and
the E (or C) index (see Table 3). The EOF2 thus captures the share of the coupled variability that is not linearly related to the
remote ENSO forcing, which is also supported by the low correlation between PC2 and the EKW time series (see Table 3).
This is independent of the periods over which the EOF analysis is performed. For the 1999-2010 period the EOF analysis (Fig.
9 d, e, f) indicates significant distinct characteristics than the EOF modes obtained for the entire period. In particular, the PC1
time series is highly correlated to the C index when the EOF analysis is performed without including the 1997-1998 El Niño
event, which indicates that the dominant EOF modes over the two periods capture the share of the coupled variability that is
related to the different El Niño types. The patterns of EOF1 for the two periods (Figs. 9 b, e) are comparable for temperature,
salinity and oxygen, that is, a strong positive loading over most of the water column. Differences can be observed at the surface
for oxygen, where the loading of the EOF1 mode pattern over 1996-2010 is weaker near the surface compared to the period
1999-2010. This could be related to the strong warming of the mixed-layer during the 1997-1998 El Niño event that leads to
reduced solubility of oxygen and compensates for the downwelling-induced oxygenation. Differences in mode patterns
between the two periods are more pronounced for nitrate and to a lesser extent for nitrite. In particular, the nitrate profile
exhibits a bend near 30m for the period 1996-2010 that is more marked than for the period 1999-2010. Differences in mode
patterns between the two periods are emphasized for mode EOF2 (Figs. 9 c, f), which is difficult to interpret. Note that the
oxygen profiles have an opposite sign between the two periods, which suggests a completely different coupled dynamics
associated to the "natural" mode of variability. Overall the results of the EOF analysis suggest two different regimes of the
coupling between physics and biogeochemistry over the two periods, which is associated to the El Niño types. A "strong"
regime associated to the strong 1997-1998 EP El Niño event and a "weak" regime corresponding to the subsequent period
where the environmental forcing consists in the alternation of CP El Niño events and La Niña events and the enhanced EKW
activity. The coupling efficiency (i.e. "strong" versus "weak") is provided by the amplitude of the mode patterns (Figs 9 b,e),
and to some extent by the percentage of variance of EOF1, 43% for the period 1996-2010 and 37% for the period 1999-2010.

The EOF analysis also indicate that the "forced" (i.e. by ENSO) coupled variability (EOF1) is related to ENSO

through the forcing of the second baroclinic mode Kelvin wave as evidenced by the large positive correlation between EKW_2
and PC1 for the two periods (Table 3). The EOF2 mode does not exhibit any linear relationship with ENSO and may account
for the large natural variability of the coastal system and/or higher-frequency environmental forcing (e.g. internal waves, intra-
seasonal variability from oceanic or atmospheric origin) that can rectify on the circulation at inter-annual time scales. For
instance, intra-seasonal Kelvin wave forcing may trigger changes in the circulation along the coast (Illig et al., 2014) that
subsequently acts on the biogeochemistry through non-linear dynamics (Vergara et al., 2016). Due to limitations in the data
set, it is difficult to address this issue that would require the experimentation with a regional coupled model. This is beyond
the scope of the present study.

**4 Discussion and concluding remarks**

The data of temperature, salinity, oxygen and nutrients between 1996 and 2011 in the central area of the Peruvian upwelling
system reveal a rich spectrum of variability. The record encompasses one of the few strong Equatorial Pacific (EP) El Nino
events observed over the last five decades and the series of Central Pacific (CP) El Nino events of the 2000s, which allows
documenting the OMZ dynamics under two different ENSO regimes and thus extends a previous study (Gutiérrez et al., 2008).
Our analysis reveals two contrasting biogeochemical regimes associated to the two El Niño types. During "strong" regimen
(strong 1997-1998 EP El Nino events), the biogeochemical properties are largely constrained by the wave-induced
downwelling conditions reflected on extreme oxygenation, reduced nutrients availability and decrease nitrogen lost processes
(denitrification, anammox). During the "weak" (CP El Niño events), less intense downwelling conditions determine a less
intense OMZ (oxygen concentration increase weakly), higher nitrate concentration and nitrogen lost processes appear not to
be significant. While under the 1997-1998 EP El Niño the biogeochemical activity was clearly coupled to the physical forcing,
this was not evident during the dominant CP El Niño regime. This reflects the distinct oceanic teleconnection through the
equatorial Kelvin waves in which mean upwelling conditions are hardly altered during the CP El Nino events (Dewitte et al.,
2012). The interpretation of the variability in these two regimes is consistent with the results of our EOF analysis combining
physical and biogeochemical data (Figure 9) which indicates that the statistics (explained variances of the modes and mode
patterns) are modified whether or not the 1997-1998 El Niño event is considered in the analysis. In particular, whereas the
dominant EOF mode that relates to the ENSO remote forcing explains less variance when the 1997-1998 El Niño is not
considered in the analysis, the second EOF mode that is independent of ENSO, exhibits a drastic change in the mode patterns
for oxygen, nitrite and nitrate. This second mode can be interpreted as resulting from the natural variability of the coupled
system, that is, the variability associated to non-linear processes in the biogeochemical system or to residual effect of other
oceanic processes (e.g. eddy activity) on the mean circulation (i.e. rectification processes). The existence of a "natural"
variability in the OMZ is suggested by regional model simulations (Bettencourt et al., 2015; Vergara et al., 2016). Our results
propose that such natural variability would be larger over periods when the frequency of CP El Niño events occurrence would
be larger than the one of EP El Niño event.

The data set also offers the opportunity to explore longer timescales of variability. Our analysis in particular suggests

a negative trend in oxygen concentration associated to a warming over the period 1999-2011. This is in contrast with the long-
term deoxygenation trend over the last decades in the eastern tropical Pacific by Stramma et al. (2008; 2010). This suggests
that either the low-frequency oxygen variability off Callao is not representative of the low-frequency changes of the off-shore
OMZ or that the trend in our data is associated to decadal changes (since it is estimated over only 13 years). This would deserve
further investigation in order to reconcile observations in the off-shore ocean and at the coast off Peru. The other striking
feature in our data set is that only temperature and oxygen experience a significant trend from 1999 (See Table 2). While the
interpretation of the latter remains uncertain, it is consistent with the existence of a significant natural variability since trend
could also emerge from non-linear processes embedded into the biogeochemical coupled system. The better understanding of
the natural variability would certainly benefit from the experimentation with a regional coupled model, which need take in
account for future work.

We now discuss some limitations of our analysis. First, our interpretation of the OMZ variability does not consider

aspects of the wind forcing, although the latter is highly variable in the central Peru region and is influential on the upwelling
dynamics. While during El Niño events, there is, in general a weakening of the upwelling favorable winds at regional scale
due to the relaxation of the South Eastern branch of the trade winds, near the coast winds can intensify locally because of
underlying warm waters effect (Dewitte and Takahashi, 2017). To which extent such anomalous winds influence the local
oceanic circulation and associated biogeochemical response remains to be investigated. Considering the limited knowledge on
this aspect and limitations in the wind data sets (Goubanova et al., 2011), we have not introduced the analysis of the local wind
forcing at inter-annual timescale here. However, this issue certainly deserves further investigation. Another important
limitation of our study is associated to the sampling of the data. The latter may result in an aliasing of the high-frequency
fluctuations embedded into the environmental forcing (i.e. intra-seasonal Kelvin wave and winds, cf. Dewitte et al. (2011))
into the low frequency fluctuations and may bias the estimate of the low-frequency mode. Gridding procedure to fill in gaps
in the data can also introduce unrealistic timescales of variability. We have here confronted two gridding procedures (one
based on simple low-pass filtering and other using a 6-month running mean filter) and they qualitatively lead to comparable
results in terms of the inter-annual and long-term variability. It would be interesting to quantitatively assess the effect of the
aliasing of high frequencies onto the monthly mean based on observations. The use of a long-term regional coupled model
simulation would be also valuable for addressing this issue. The latter could in particular allows investigating the remote
oceanic forcing associated to the intra-seasonal Kelvin wave on the oxygen conditions off Peru, which was not possible from
our data set. It could also help in better understanding how the intra-seasonal variability can rectify on the inter-annual
variability through non-linear processes (e.g. eddy transport). Despite these limitations, the results presented in this paper are
valuable as a benchmark for the validation of regional coupled models that are aimed to address the OMZ dynamics.
Overall, our results illustrate the rich spectrum of OMZ variability at inter-annual timescales, which cannot be solely
interpreted as resulting from the ENSO oceanic teleconnection. Natural variability in the OMZ is expected from the complex
of processes involved. Our analysis provides a first assessment of such a variability from observations. Understand the intensity
and distribution of the OMZ is essential to understand changes in nutriments and finally to predict the productivity and
distribution of marine resources.  The existence of the two regimes suggested here need to be tested from global or regional
models, which could provide a pathway for understanding the sensitivity of the OMZ to climate variability and for a better
prediction of global change scenarios.

*Acknowledgements*
This research was supported by the Instituto del Mar del Peru (IMARPE). We thank Carlos Robles and Miguel Sarmiento, the
technical chemical staff. Thanks to the crew of the IMARPE VIII and the SNP-2 research vessels and all the scientific
colleagues that help us. This work is a contribution of the project "Integrated Study of the Upwelling system off Peru"
developed by the first author in the Direction of Oceanography and Climate Change Research of IMARPE. Boris Dewitte
acknowledges support from FONDECYT (project 1151185) and IRD. Mercator is thanked for providing the model data to
derive the Kelvin wave estimate. We are grateful to the two anonymous reviewers for their constructive comments that helped
to improve the manuscript. This work is a contribution of the project "Integrated Study of the Upwelling system off Peru"
developed in the Direction of Oceanography and Climate Change Research of IMARPE.

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

**Table 1: El Niño and La Niña years and their characteristics (magnitude and type)**

| El Niño (red) and La Niña (blue) years | Magnitude (ONI index) | Type (Eastern Pacific -EP versus Central Pacific-CP) (Yu and Kim, 2013) |
|---|---|---|
| 1997/1998 | Strong (extraordinary) | EP |
| 1998/1999 | moderate | |
| 1999/2000 | moderate | |
| 2000/2001 | weak | |
| 2002/2003 | moderate | CP |
| 2004/2005 | moderate | CP |
| 2006/2007 | moderate | Mixed |
| 2007/2008 | moderate | |
| 2009/2010 | moderate | CP |
| 2010/2011 | moderate | |


**Table 2: slope of the linear fit for oxygen, temperature,salinity, nitrate and nitrite as a function of depth over the period 1999-2011.**
**The slope for thermocline and oxycline depths are also provided as a function of season. The confidence level estimated based on a**
**Student's T-test is indicated in parenthesis when larger than 80%.**

| Depth (meter) | $O_2$ ($\mu$mol kg$^{-1}$ decade$^{-1}$) | | T (°C decade$^{-1}$) | | S (PSU decade$^{-1}$) | Nitrate ($\mu$mol L$^{-1}$ decade$^{-1}$) | Nitrite ($\mu$mol L$^{-1}$ decade$^{-1}$) |
|---|---|---|---|---|---|---|---|
| 0 | 24.03 (90%) | | -0.04 | | 0.026 | 0.93 | 0.11 |
| 10 | 47.55 (95%) | | 0.53 (80%) | | 0.013 | -0.17 | -0.22 (80%) |
| 25 | 40.35 (95%) | | 0.65 (90%) | | 0.025 (80%) | -1.67 | -0.01 |
| 50 | 14.40 (85%) | | 0.50 (90%) | | 0.003 | 0.01 | -0.15 |
| 75 | 6.04 (90%) | | 0.34 (90%) | | -0.002 | 1.85 | -0.57 |
| 90 | 6.76 (95%) | | 0.42 (95%) | | -0.001 | 2.51 (80%) | -0.75 |
| 100 | 7.53 (95%) | | 0.46 (95%) | | 0.003 | 2.95 (80%) | -0.88 |
| **Annual** | **OMZ** (m decade$^{-1}$) -0.64 (95%) | | **Thermocline** (m decade$^{-1}$) -0.30 (95%) | | | | |
| **Seasonal** | **Summer** -0.74 (95%) | **Winter** -0.77 (95%) | **Summer** -0.63 (95%) | **Winter** 0.03 | | | |
| | **Fall** -0.76 (95%) | **Spring** -0.69 (95%) | **Fall** -0.49 (95%) | **Spring** -0.48 (95%) | | | |




**Table 3: Correlation values between the PC time series, the ENSO indices (E, C) and Kelvin waves amplitude (EKW 1 and 2) for**
**the periods. Shading indicates the correlation values significant at the 95% level. E is the Eastern Pacific El Niño index) and C is the**
**Central Pacific index as defined by Takahashi et al. (2011). Note that the C index accounts for both Central Pacific El Niño events**
**and La Niña events.**

| | **E** | **C** | **EKW1** | **EKW2** |
|---|---|---|---|---|
| **Period** | **1996-2010** | | | |
| **PC1** | 0.72 | 0.31 | 0.38 | 0.61 |
| **PC2** | 0.28 | -0.27 | 0.00 | 0.07 |
| **Period** | **1999-2010** | | | |
| **PC1** | 0.23 | 0.58 | 0.33 | 0.53 |
| **PC2** | -0.10 | 0.09 | 0.08 | -0.01 |



**Figures caption**
**Figure 1.** Location of the sampling station (St 4; 20 nm, 145 m depth) in the coastal upwelling ecosystem off central Peru, Callao (12º 02´
S; 77° 29´ W).
**Figure 2.** Time series of temperature (°C) (a), salinity (b) and dissolved oxygen ($\mu$mol kg$^{-1}$) (c) during the 1996-2011 study years. Black
dots indicate the location in space and time of the data and a DIVA 2017 graphic interpolation was used for the visualization of the data.
**Figure 3.** Temperature (top) and Oxygen (bottom) anomalies relative to the mean climatology (a, c) and the respective climatologies (b,d).
The 45 $\mu$mol kg$^{-1}$ iso-contour for total oxygen concentration (oxycline) is overplotted in c) as a thick black line while the 90 $\mu$mol kg$^{-1}$ iso-
contour for anomalies is drawn in thin black line in order to visualize the amplitude of the anomalies during the 1997/98 El Niño event. The
number of years to derive the climatology is indicated in b) and d) below the calendar months.
**Figure 4.** Time series of nitrate (a), nitrite (b), silicate (c) and phosphate (d) during 1996-2011. Units are $\mu$mol L$^{-1}$. Black dots represent the
data and the figure present a DIVA 2017 graphic interpolation only for the visualization of the data.
**Figure 5.** Time series of N deficit ($\mu$mol L$^{-1}$) at St. 4 off Callao during 1996-2011. Black dots represent the data and the figure present a
DIVA 2017 graphic interpolation only for the visualization of the data.
**Figure 6.** Projection of the data onto the (red line) E and (blue line) C indices for a) oxygen, b) temperature, c) salinity, d) nitrate and e)
nitrite.
**Figure 7.** Percentage of variance explained by the projection of the data on the E and C modes as a function of depth for temperature (red),
salinity (blue), oxygen (dark green), nitrate (purple) and nitrite (orange).
**Figure 8.** Evolution of the (a, b) amplitude of the Equatorial Kelvin Waves (EKW) anomalies at 90°W for the first (EKW_1) and second
(EKW_2) baroclinic modes. Units are cm (equivalent sea level). The standard deviation is indicated by the horizontal dashed lines. Time
resolution of the data is every 5 days, (c) depth of the thermocline; (d) OMZ upper boundary depth and (e) Fixed Nitrogen deficit (Ndef) at
50 m (f ) at St. 4 off Callao during 1996-2011. On the right hand side of each timeseries, the global wavelet spectrum with significance level
at 95% are shown (g-j). The 95% confidence level estimated from a red noise (Markov model).
**Figure 9.** Combined EOF analysis of temperature, salinity, oxygen, nitrate and nitrite for the periods (top) 1995-2010 and (bottom) 1999-
2010. (a, d) PC timeseries and (b, c, e, f) mode patterns for the first two EOF modes. A 6-month low-pass filter was applied to the data and
the data have been normalized prior to carrying the EOF analysis so that unit is adimensionalized.


**Figure 1.**

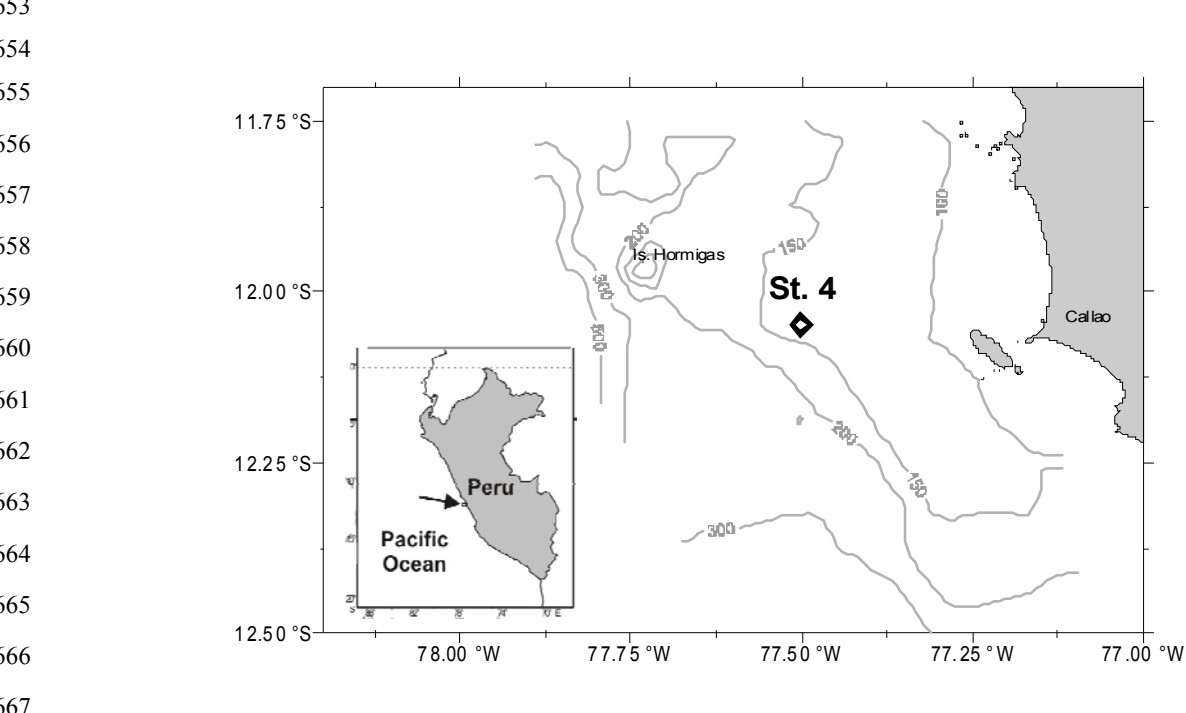





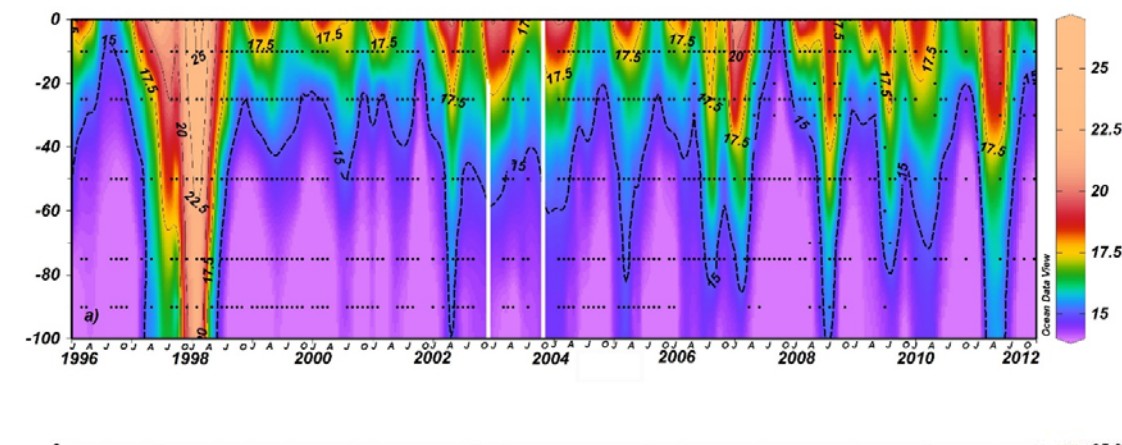

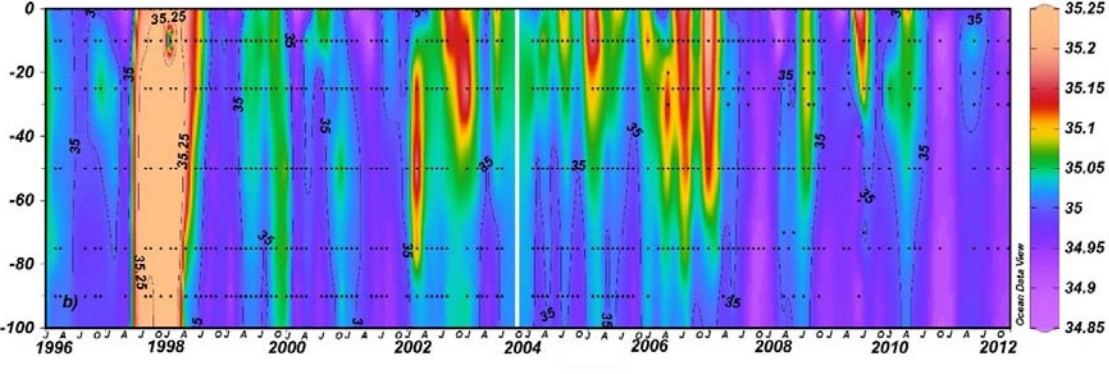

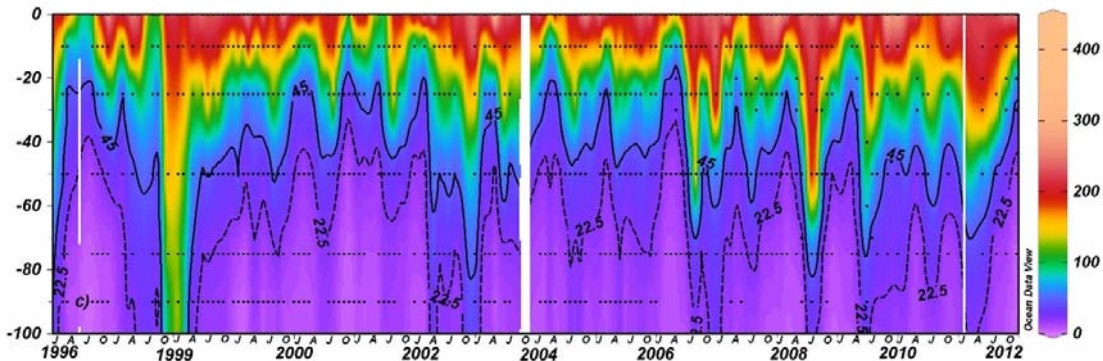

**Figure 2.**









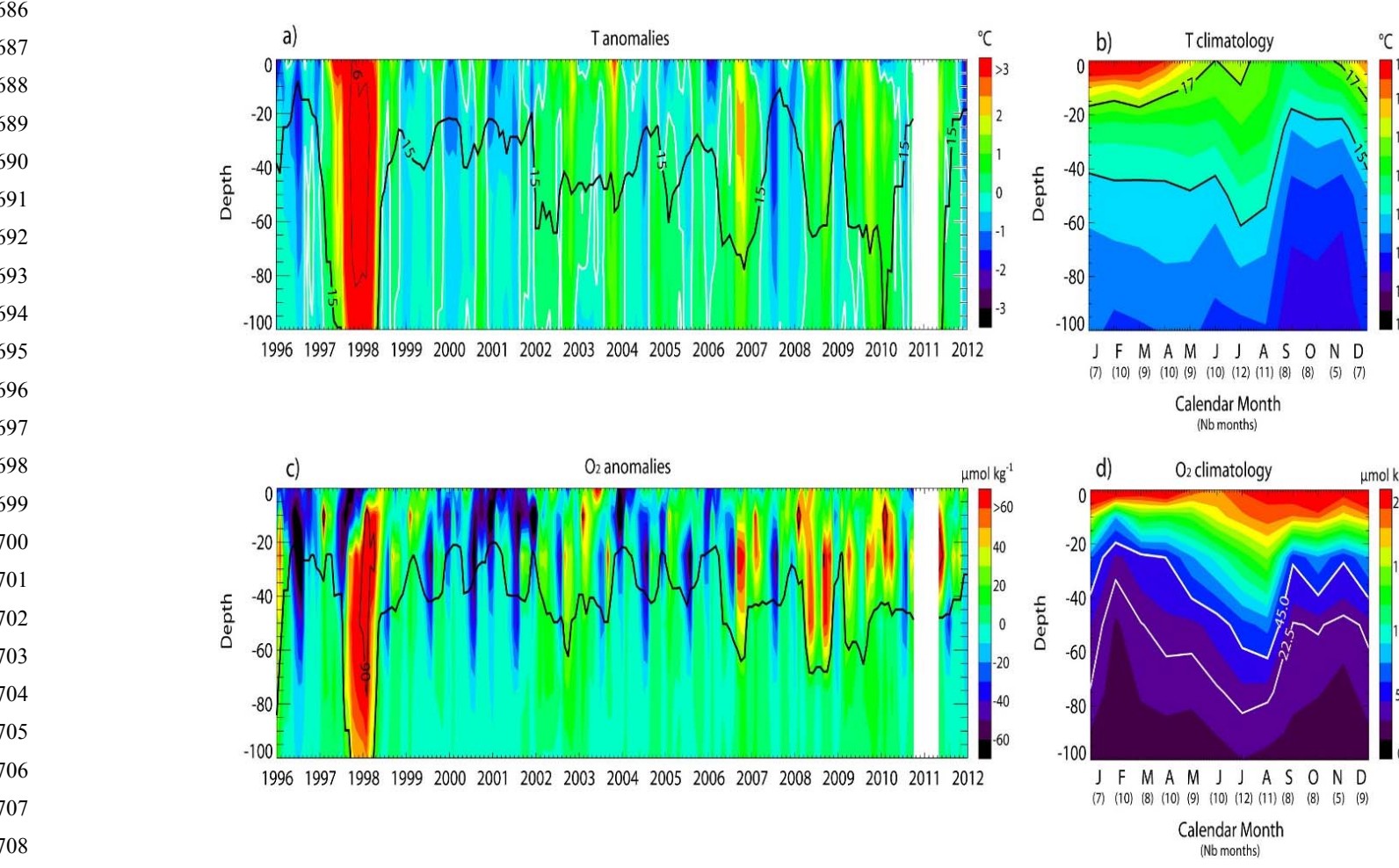

**Figure 3.**













































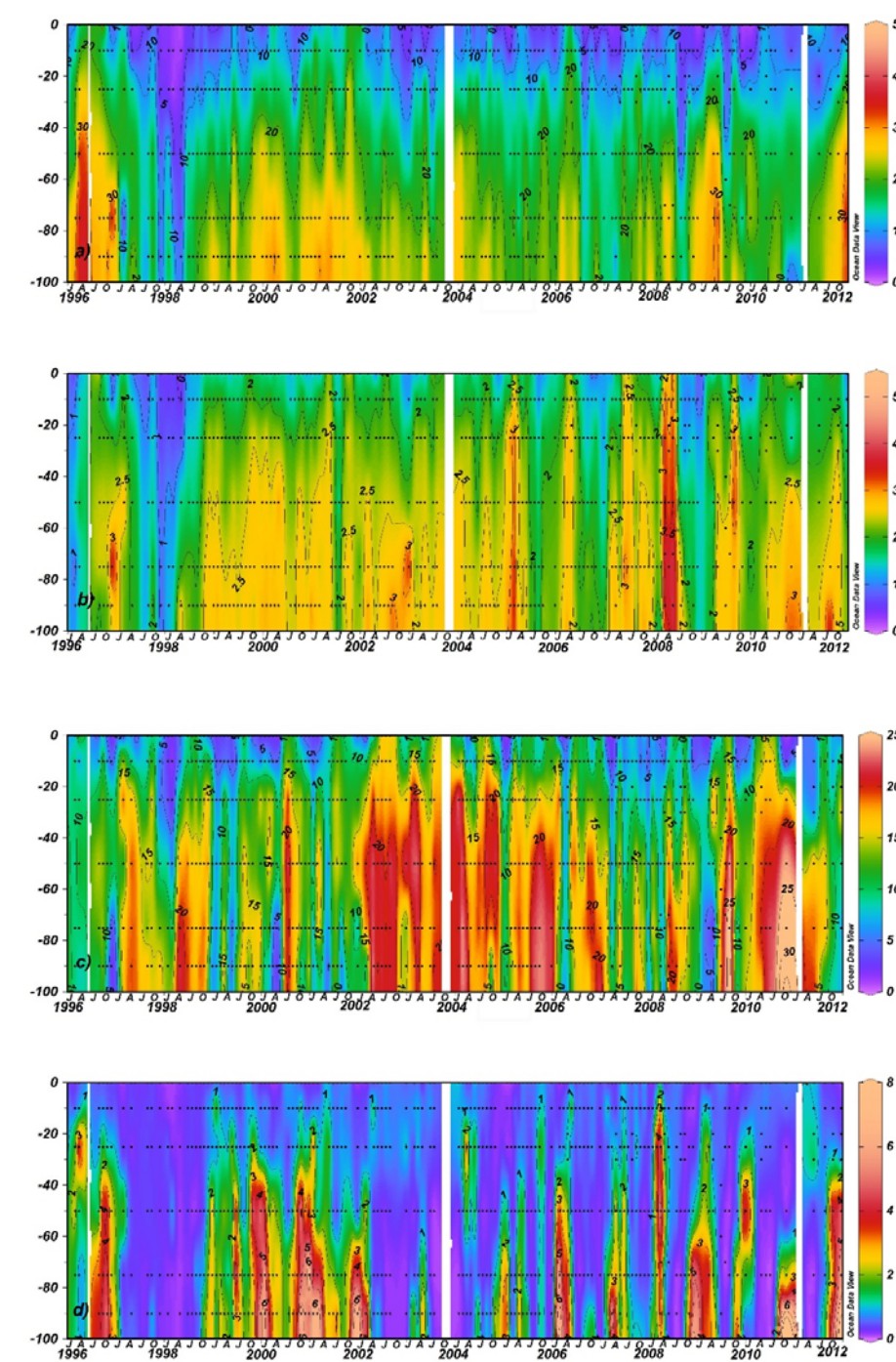

**Figure 4.**

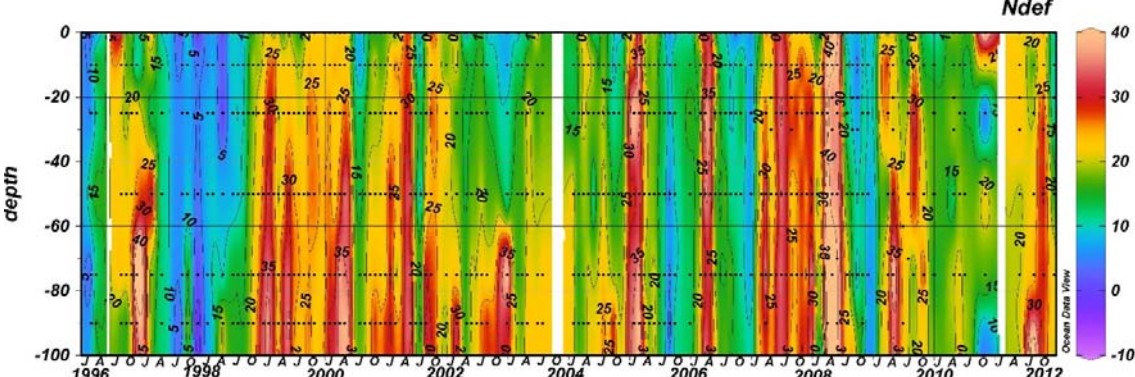

**Figure 5.**



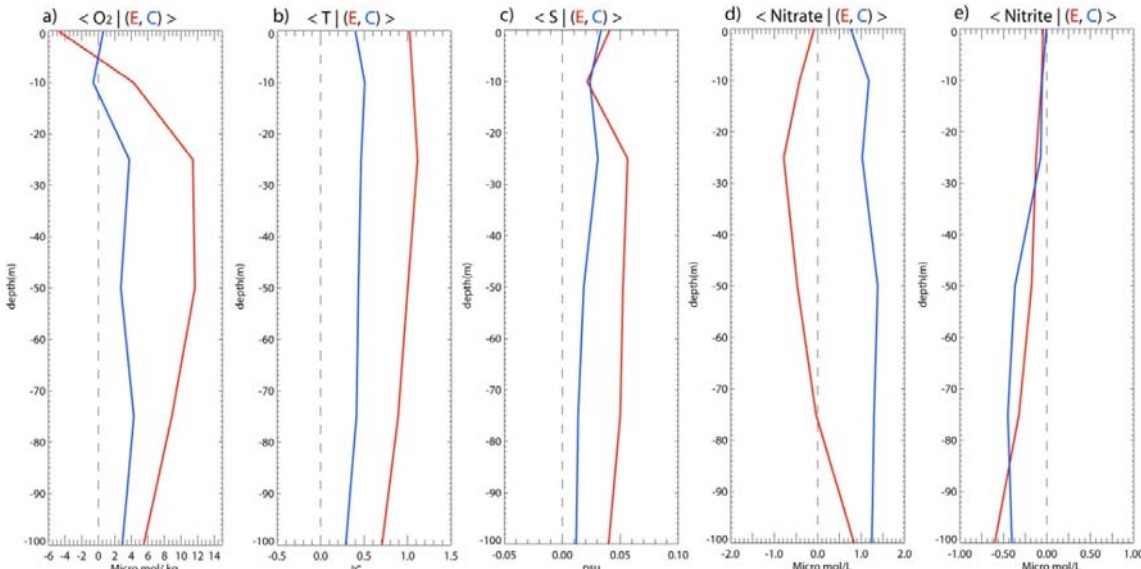







**Figure 6.**









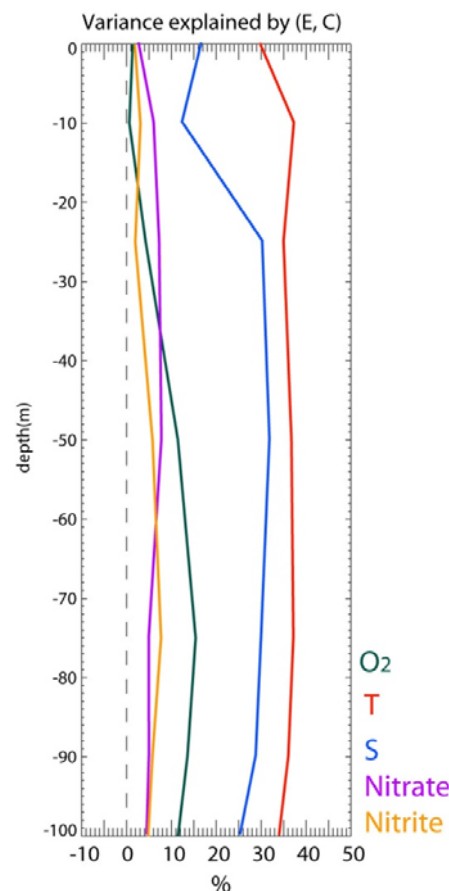


**Figure 7.**









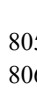

Figure 8.





















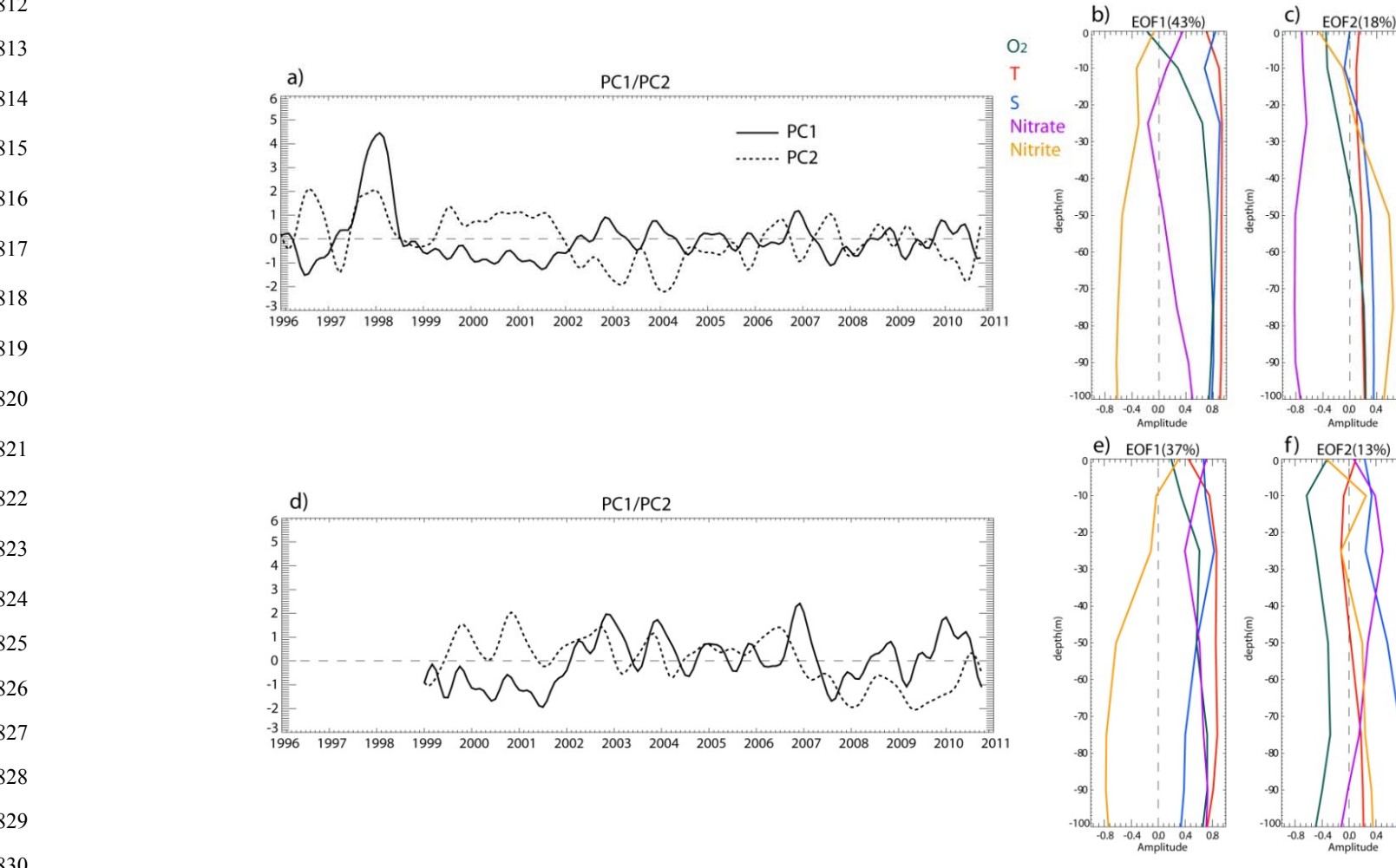

**Figure 9.**