# Peer review of "The OMZ and nutrients features as a signature of inter-annual and"

_Biogeosciences, 2015_

## Referee Comment (RC1) · Anonymous Referee #1 · 19 Feb 2016

Review of the manuscript bg-2015-567: The OMZ and nutrients features as a signature of interannual and low frequency variability off the Peruvian upwelling system

by M. Graco, S. Purca, B. Dewitte, O. Moron, J. Ledesma, G. Flores, C. Castro and D. Gutierrez

General comments: The manuscript by Graco et al. is a combination of a lengthy literature review and the discussion of long time-series off Lima. The time series are great and some background information from literature is helpful, however in the present version the manuscript is difficult to read and needs to be modified. What makes the manuscript difficult to read is the bad English (I know all authors are no native English speakers, but some help should be searched to make the manuscript better readable),

the lengthy literature review with a lot of old and 'grey literature' while the scientific results are not well enough described. IEKW are a main subject discussed in the text, however the information on IEKW is too short. E.g. in paragraph 2.4 (page 7) it is stated that sea level height at 90°W was used to derive IEKW. No information on the region used is explained, was it e.g. 90°N to 90°S, or some selected region? The figures are not called in increasing order, Figure 6a (page 8 line 22) is called before figure 3. The presentation of wrong links (see below) gives the impression that none of the co-authors read the text carefully and they decided to leave their job to the reviewers. If possible, an extension of the time series at least until the end of 2011 would improve the manuscript considerably (see below). I recommend major revisions of the manuscript.

Specific comments: As mentioned above, the time series are great. In the text it is stated, that the records are used until the end of 2009. No information is given, that the measurement program stopped. In case the measurements continued until the end of 2011 it would improve the manuscript considerably to extend the time series. The time series presented in the manuscript include a strong El Nino in 1997/1998. With regard to La Nina a moderate event existed in 2007/2008, which is visible in the temperature and salinity record but not well developed in oxygen and nutrients. From July 2010 to April 2011 a strong La Nina existed, and this would allow to compare the signal of a strong La Nina with the strong El Nino as well as with the moderate La Nina if the time series can be extended to the end of 2011.

In the manuscript relative large interannual/intraseasonal variability since 2002 is discussed. However the time period covered before 2002 is short and strongly influenced by an El Nino and a La Nina event, hence a longer time series would be needed for a clear statement. In addition, it is mentioned that in 2002 the measurements changed from Niskin bottles to CTD measurements, hence a discussion how this change in method influences the measurements is needed.

Three examples that give the impression that the co-authors did not read carefully the

manuscript: 1) Page 8 line 19, page 10, line 28 and page 11 line 7: '(see sect. 3.5)'. There is no section 3.5 in the manuscript.

2) Page 13 line 3: '(Fig. 8f and g)'. There is no Figure 8f and g.

3) Ndef is defined different in the text (page 6 line 2) and the figure legend of figure 7. In the abstract lines 12-13 Intraseasonal should be written directly before Equatorial Kelvin Wave that the definition (IEKW) makes sense. Abstract lines 13-15. What do you mean with 'increase of the IEKW'? I can't see it in figure 8.

Normally an abbreviation is defined and then used in the following text. In this manuscript the repeated definitions of the same abbreviations are strange and make the manuscript difficult to read. E.g. IEKW is defined on page 7 two times in lines 1 and 2, on page 8 line 18, page 11 line 27 and page 17 line 1.

ESSW is defined page 3 line 8, page 7 line 25 and page 9 line 10. Similar for other abbreviations.

Figure references should make sense. Page 9 line 15: (O2 < 8.9 micromol/kg, Fig.3) has nothing to do with Figure 3.

The units used in the figures and text should be the same. Oxygen in the text is presented as micromole/kg (e.g. page 9 line 8), in figure 3 as micromole/L; for nutrients in micromole/L (e.g. page 10, line 14) in Figure 5 as microM.

Statement on page 9 lines 10 to 12 is difficult to see in figure 4b and 4c and leaves only winter as deeper OMZ boundary. May be use months instead of seasons to better define the period you refer to.

Page 10 line 28 to page 11 line 2: The changes mentioned are difficult to see in Figure 5d, may be add the depths range in the text you are considering.

Page 10 lines 3 to 5. PC1 And PC2 are mentioned and shown in Figure 6b. In paragraph 2.3 (page 6) where the method is described, only PC1 is mentioned. PC2 should

be also explained in paragraph 2.3.

The statement page 18 line 29 to page 19 line 2 either needs a reference if related to literature or a better explanation if derived here.

Technical corrections:

Page 3 line 13: Gutierrez et al 2011 is a and b in the reference list, which one?

Page 3 line 20 Pacific, not pacific

Page 7 lines 21-22 'considered' instead of 'considering'

Page 8 line 5: Better define 'Z_15' on page 7 line 21. On page 12 line 24 a 'Z_15°C' is defined, I guess this is the same as 'Z_15' hence don't introduce another abbreviation for the same thing.

Page 8 line 1: 'isopycnal' not 'isopycn'.

Page 8 line 15 or in the legend of figure 4 write like in line 12 '15°C isotherm' not only '15° isotherm'

Page 10 line 15 does not make sense. 'and' instead of 'an'?

Page 14 line 14: Criales et al., 2006, there is only a reference Criales-Hernandez et al., 2006 in the reference list.

Page 17 line 5: 'We proposed. . .' you do it here hence write 'We propose. . .'

Page 17 lines 24 to 29: A long sentence which does not make sense. It would be ok, if you remove 'evidence' and 'that' in line 25, is that what you like to say?

Page 18 line 21: Morales et al. 1996 is not in the reference list.

Page 18 line 22: Thamdrup and Dalsgaard, 2002 is not in the reference list.

Page 20 line 18: '. . .data out field work' remove 'out'

Page 21 line 19: 'California Current', not 'California'

Page 24 line 28: 'Ehrhardt', not 'Ehrdardt' I only know a 1999 version of this book with Kremling as second author and Ehrhardt as third author and Anderson only listed as one of many contribution authors but not in the book author list. Please check, whether the version you listed also exists.

Figure 1: Geographical coordinates in the inset are not readable. In the figure legend it is stated 12°S, in the text 12°02'S. As the station in the figure is plotted south of 12°S better write also 12°02'S in the figure legend.

Figure 2: The numbers in the figures are not readable in the BGD version. For a final version either the figure or the numbers need to be larger.

Figure 3: Please provide the unit for oxygen in Fig. 3a either at the color bar or in the figure legend. In Figure 3b 44.6 and 22.3 are described in the figure legend but 45 and 22.5 are shown in the figure. This needs to be the same!

Figure 5: The figure is too small to read the numbers in the figure. The figure legend states that the 22.3 and 44.6 oxylines are included. I can't see them either they are not included or the figure is too small to locate them.

Figure 6: Include 'EOF' in the figure legend.

Figure 7: Remove 'Time (station date&time) [years since 0000-01-01]' from the figure and give the unit for Ndef in the figure legend (micromole/L used in the text for Ndef).

Figure 8 legend: 'On the left, different variables...' I guess it should read "On the right..." By the way the axis on the right of figure 8 are too small and unreadable.

---

## Referee Comment (RC2) · Anonymous Referee #2 · 4 Apr 2016

The paper addresses an interesting topic of physical-chemical coupling in the northern Humboldt System. This region is particular attractive because of its strong connection with equatorial dynamics, high biological productivity, as well as a shallow and intense oxygen minimum zone that influences nitrogen cycling and biological dynamics. The study examines the temporal and vertical variability of relevant physical and chemical variables during the period 1996-2009, covering the strong El Nino 1997-98 and subsequent La Nina 1998-99, as well as period with relatively neutral or weak ENSO conditions after 2002. The authors conclude that most of the physical-chemical variability off Peru is linked to equatorially originating remote forcing. They also suggest enhanced physical-chemical intraseasonal variability after 2002, associated with

a change in the intraseasonal equatorial Kelvin wave (IEKW) activity. None of those statements is well supported in the paper. Three aspects could be argued against the conclusions. First, since the local forcing (e.g. coastal winds, heat fluxes) was not examined, it is not possible to conclude that remote forcing is the dominant forcing. Second, the coastal time series have monthly resolution that precludes a proper characterization of the dominant 40-60 day intraseasonal variability off Peru, making difficult to connect the physical-chemical changes off Callao to the IEKW variability. Third, the characterization is strongly dependent on El Nino 1997-98 and subsequent La Nina disturbances. It is not clear for me if a really novel scenario exists after 2002.

The paper has the potential to address a very interesting topic, as the interannual to intraseasonal variability of chemical variables remains not well documented off Peru. However, there are many aspects that need revision, including goals and methodological issues. The paper structure and writing need substantially improvement.

Specific issues that need revision:

1. Introduction

The introduction is weak. I suggest be more concise and define better the paper goals. The sentence "in order to infer potential biogeochemical scenarios in connection with equatorial variability" appears to me too vague.

2. Methods 2.1 Sentences from lines 8 to 13 are not needed in this section.

2.2 The instruments to measure temperature and salinity changed in 2002. Could that explain the apparent change in the physical-chemical coupling after 2002? Do you have a period with overlapping observations of thermometer, salinometer, and CTD to check the measurements consistency?

2.3 Is the EOF analysis really needed? Only the PC time series were shown, what about the spatial pattern?

3. Results

I suggest Figure 2 be a 8 panels figure, with the time-depth diagrams of temperature, salinity, density, and oxygen in the left panels, and the corresponding monthly climatological mean in the right panels. In that way you could eliminate Figs. 3 and 4, and you do not need to go back and forward in the figure's citation.

3.1 Page 8 Line 17: Year 2008 is not El Nino according to El Nino 3.4 index. Lines 22-24: How do you support the $Z\_15°C$ - ENSO connection?

3.2 Page 9 Line 21-23. I do not think it is possible to discriminate properly the intraseasonal from the seasonal variability with monthly observations.

Why R-square instead of r when reporting correlation coefficients? That appears to be a conceptual error

3.3 Page 10 Line 10-12. Maximum NO3 in winter could be linked to decreased plankton uptake due to light-limitation (Echevin et al., 2004)

Why phosphate is not limiting? Need to support it

Page 11 The connection between PC2 and the intraseasonal variability is not clear for me. Could you quantify the connection?

3.4 Is it possible to include numerical model outputs in the analysis? Several regional model efforts have been done in the region that might be useful to interpret better the observations.

I do not understand why do you use only the global wavelet spectrum. Examining wavelets in the time-frequency domain is more informative, especially to see temporal changes in the IEKW activity.

Where is Section 3.5? Cited three times in the text

4. Discussion

Overall the discussion needs a better integration of biogeochemical processes influencing the observed vertical profiles of oxygen and nutrients.

Page 16 The statement that 84% of variability could be remotely forced is totally unsupported. You get those percentages from the EOF analysis, but the link between equatorial variability and El Callao series was not quantified. What about local winds? Wind stress curl?

Same thing, that the leading EOF of the chemical components explains 50% does not mean that 50% of the variability is linked to remote forcing.

―――――――――――――――

---

## Author Comment (AC1) · 9 Jun 2016

Dear reviewers,

We highly appreciate all the comments and suggestions from both reviewers about our manuscript entitled "THE OMZ AND NUTRIENT FEATURES AS A SIGNATURE OF INTERANNUAL AND LOW FREQUENCY VARIABILITY OFF THE PERUVIAN UP-WELLING SYSTEM" by Michelle Graco, Sara Purca, Boris Dewitte, Octavio Morón, Georgina Flores, Jesús Ledesma, Carmen G. Castro, and Dimitri Gutiérrez (This manuscript is for the Special Issue: Biogeochemical processes, tropospheric chemistry and interactions across the ocean/atmosphere interface in the coastal upwelling off Peru).

[Figure]

The manuscript is being revised following the reviewers' suggestions, which are greatly improving it. Based on their comments, we are rebuilding the entire time series expanding them till December 2011. This task is taking more time than expected and consequently we are still working on the revised manuscript. In the following paragraphs, we are answering all reviewers' comments, describing in detail the way we are proceeding. We really appreciate if you could consider them to continue working on the modified version of our manuscript.

We agree that expanding the time series still 2011 will reinforce our results and clarify the interannual/ intraseasonal variability as both reviewers indicate. In this way, we would also cover La Niña 2010-11 event, as indicated by reviewer 1. In the section of results we present some details about that.

Regarding the impact of local winds on the intraseasonal variability, we completely agree with reviewer 2 's suggestion about including it in our discussion. We have expanded the discussion on the role of wind forcing based on previous studies (Dewitte et al 2011, Echevin et al 2014; Illig et al., 2014) (see discussion section). In our study we are not addressing the higher-frequency variability associated to wind forcing owed to the limitations of the data sets (monthly resolution). We now mention such a limitation of our study in the revised manuscript.

Regarding the reviewer 2 comment that the characterization is strongly dependent on El Nino 1997-98 and subsequent La Nina disturbances and the novel scenario after 2002. We have toned down the idea that two different scenario exists from before and after 2002, and rather contrast the response of biology to two different types of events, a strong Eastern Pacific El Niño event associated to a radical change in the regional hydrological conditions, and Central Pacific El Niño events that are associated to a significant high-frequency remote forcing under relatively mean normal conditions (see Dewitte et al. (2012)).

We have thoroughly checked over the grammar and syntax, and a native English

speaker will examined the revised manuscript. The paper is being shortened and reshaped to state explicitly the main goals of the manuscript. Thank you for all the specific comments that will be taken into in order to improve each section.

1- About the INTRODUCTION:

We agree with reviewer 2 that is necessary a more concise and better definition of manuscript objectives.

2- About METHODOLOGY:

a-Following the reviewer's 1 and 2 recommendation, we rewritten the description of the method to derive the Kelvin wave. The new paragraph 2.4 writes as follows.

The amplitude of the Intraseasonal Equatorial Kelvin Wave (IEKW) is derived from the SODA oceanic Reanalysis (Carton and Giese, 2008). The method consists in projecting the pressure and current anomalies from SODA between 15°S and 15°N onto the theoretical vertical mode functions obtained from the vertical mode decomposition of the mean stratification. Kelvin wave amplitude is then obtained by projecting the results onto the horizontal modes at each grid point in longitude. The method has been shown to be successful in separating first and second baroclinic waves (Dewitte et al., 1999, 2008) that propagate a different phase speed and impact the Peru coast in a very specific way (Illig et al., 2014). In particular due to the sloping thermocline from west to east along the equator, the second baroclinic mode Kelvin wave is more energetic and influential on the upwelling variability off the Peruvian coast (Dewitte et al., 2011, 2012). For the correlation analysis with the dissolved oxygen data, we select the IEKW amplitude at 90°W.

To extend the IEKW for 2011, we will use the outputs of a general circulation model forced by an atmospheric Reanalysis. This simulation belongs to the set of MERCA-TOR global OGCM simulations (http://www.mercator-ocean.fr/) was used in Mosquera et al. (2014) to estimate the Kelvin wave amplitude.

References added to the revised manuscript:

Illig S., B. Dewitte, K. Goubanova, G. Cambon, J. Boucharel, F. Monetti, C. Romero, S. Purca and R. Flores, 2014: Forcing mechanisms of intraseasonal SST variability off Peru in 2000-2008: local versus remote forcings. J. Geophys. Res.-Oceans, Vol. 119, 6, 3548-3573. Mosquera-Vásquez, B. Dewitte and Serena Illig. The central Pacific El Niño intraseasonal kelvin wave JGR: Oceans (2014). 10.1002/2014JC10044.

b- About the change of instrumentation, we perform comparative analysis between CTD and salinometer analysis during all the monthly cruises in order to present consistent information. We will include a discussion about that in this section as the reviewer 1 recommend.

c- Regarding the EOF analysis the reviewer 2 ask if is really needed PC1 and PC2 if only the PC1 time series were shown.

The EOF analysis is performed in order to provide a synthetic view of the variability for physical and biogeochemical parameters. We focus on the first two EOF, in the case of the physical the PC1 explained up to 84.5 % of the variance but in the chemical parameters the PC1 explained 50% and the PC2 30%. We now provide both the eigenvalue (timeseries) and eigenvectors (vertical profiles) of the EOF analysis, and also the EOF analysis combining all the variables (physic, chemistry) and also the Kelvin Wave data. We can include more detail when we present the EOF analysis, fig. 6. We have expanded the description of the results and the discussion.

d- About the spatial pattern, to answer the reviewer 2, in the study area we present the study site. A station located in the continental shelf in front of Callao at 20 nm. Previous studies in the same area but in a more coastal area at 8 nm show also the influence of the remote forcing in the benthic community and the environmental conditions particularly during El Niño.

e- The reviewer 2 indicate that the wavelets in the time-frequency domain could be

more informative that the global wavelet spectrum.

We perform an analyze in the time frequency wavelet frequency in addition to the global wavelet spectrum and finally we decide to use the GWS because both present the same information for the time series indicate the strong interannual signature associated with El Niño and also the intraseasonal signature that appear in the band of 180 days (6 months)-90 days (3 months). In the Figure 8 The GWS showed the significance interannual periods for all-time series, over the dotted line.

About RESULTS:

a- Following the reviewer 1 recommendation we have extended the period for the observations. Unfortunately we were not able to extend the series for the Intraseasonal Kelvin Wave since our estimate is based on an oceanic Reanalysis that is only available until Dec 2008. After 2008 will be necessary to use other product and perform comparative analysis.

We have also extended the IEKW timeseries until 2011. To do so, we used the outputs of a general circulation model forced by an atmospheric Reanalysis. This simulation (referred to as Mercator hereafter) was also used in Mosquera et al. (2014) to estimate the Kelvin wave amplitude. Although it is not an assimilation product like SODA, the simulation is realistic. We will also include the correlation between the Kelvin wave amplitude at 90°W for SODA and Mercator the first and second baroclinic modes, respectively.

b- We agree with reviewer's statement that the year 2008 with an El Niño 3.4 index does not indicate a warm event. However the costal El Niño index 1+2 indicate a warm period. The Peruvian coast is one of the few regions in the world that requires two indexes for monitoring El Niño. In 2012, the national technical committee for El Niño study (ENFEN; http://www.met.igp.gob.pe/variabclim/enfen/) defined the ICEN ( Coastal El Niño index) based on the anomaly of the sea surface temperature of the El Niño region 1+2 (90° W- 80°W, 10°S-0°). The ICEN index is better indicator of the

ENSO cycle off the Peruvian coast. It gives an idea not only of the El Niño impact on the physical and chemical fields but also on the El Niño consequences on the biota and consequently on the economic resources. We will include this in the methodology.

We will also extend the discussion indicating that after the 1997/98 El Niño, the interannual variability in the equatorial Pacific consists in a different type of El Niño events. Those events are referred to as Central Pacific El Niño events and are characterized by an increased variability of the intraseasonal Kelvin wave activity during the development and peak phase compared to the strong El Niño events (See Mosquera et al. (2014). Therefore the period 2000-2008 can be considered as a period with enhanced intraseasonal variability compared to the previous decade.

c- As reviewer 2 recommend we will explore the regional models outputs in order to compare our results of the IEKW.

We agree that model experiments could provide additional material for interpreting the observations. For OMZ regional modeling, we are only aware of the study by Montes et al. (2014) that addresses the seasonal cycle, not the interannual variability. It is beyond the scope of the present study to present model simulations over the period of interest, which would deserve a thorough validation. This is actually a work in progress and the results will be reported later.

d- As reviewer 2 recommended to show the quantification of connection between PC2 and the intrasesonal variability.

Page 16: The equatorial forcing the physical and chemical modes of variability is less than 60%. The linear correlation (r coefficient) between IEKW_1 and IEKW_2 and PC1 physical time series are -0.35 and -0.67, respectively and also significant correlations between the IEKW_1 and 2 with the EOF1 (PC1) for the chemistry (-0.29, -0.52 respectively) and with the oxygen minimum zone upper layer position (-0.28, -0.55) were significant during the study period. We will include a table with the lineal correlations between the mode 1 and 2 of the IEKW and the PC1 and 2 of the physical and chemical

time series. We will also explain in more details this results.

e- The reviewer 2 consider that t is not possible to discriminate properly the intraseasonal from the seasonal variability with monthly observations.

We agree with the reviewer that the monthly resolution will tend to damp the amplitude of the intraseasonal variability. However since we focus on the frequency band [40-90] days, we believe it is still reasonable to address the intraseasonal variability with the monthly data. We now will expand the discussion about this and mention in the text limitation associated to aliasing that would require to be investigated further based on mooring data for instance. There is however no biogeochemical data from a mooring available in this region. This is actually plans that IMARPE and the international community (TPOS2020 program) have.

All the specific comments and minor changes will be taken into consideration in order to improve this section.

6. About DISCUSION

In accord with the reviewer 2, we agree to include a better discussion about the biogeochemical process that significantly impact in the nutrients and oxygen profiles.

As recomended reviewer 1 We will extend the discussion indicating that after the 1997/98 El Niño, the interannual variability in the equatorial Pacific consists in a different type of El Niño events and associated with those events an increased variability of the intraseasonal Kelvin wave activity during the development and peak phase compared to the strong El Niño events (See Mosquera et al. (2014).

The reviewer 2 also recommended to discuss the local effect of Winds on the intraseasonal signature. We recognized that this is an important point to include. We have expanded the discussion on the role of wind forcing based on previous studies. In particular, Dewitte et al. (2011) shows that there exist two distinct regimes of intraseasonal variability for SST and winds along the coast of Peru: One regime with variability

timescales in the frequency band [2-30 days] and another one centered on the peak frequency of 50 days-1. While the former one is associated to local wind forcing, the latter is due to the remote equatorial forcing. Here we focus on the intraseasonal variability associated to the equatorial Kelvin wave that is the frequency band of [30-90] days that the data (model and in situ) can resolve. Echevin et al. (2014) shows evidence that subsurface nutrient and chlorophyll intraseasonal variability are mainly forced by the coastally trapped waves triggered by intraseasonal equatorial Kelvin waves reaching the South-American coast. In the northern part of the Peru shelf (latitudes 4°– 8°S) on [50–90] day time scales the authors show that the impact of the local wind-forced intraseasonal variability on the ecosystem is of a similar order of magnitude to that remotely forced. In the central and southern part of Peru, that include our study area (Callao 12°S), IEKW-forced CTW signature emerges as dominant over the local wind impact.

In our study we are not addressing the higher-frequency variability associated to wind forcing owed to the limitations of the data sets (monthly resolution). We now mention such a limitation of our study in the revised manuscript.

All the specific comments and minor changes will be taken into consideration in order to improve this section.

---

## Author Response (AR1)

Lima, the 12th of September, 2016

Michelle Graco
Department of Oceanography
Instituto del Mar del Perú (Imarpe)
Esq. Gamarra y Gral Valle s/n
Chucuito-Callao, Peru
Phone/Fax: 0051-1-429 6069
e-mail: mgraco@imarpe.gob.pe

Dear Editor,

We highly appreciated all the comments and suggestions from both reviewers about our manuscript entitled "THE OMZ AND NUTRIENT FEATURES AS A SIGNATURE OF INTERANNUAL AND LOW FREQUENCY VARIABILITY OFF THE PERUVIAN UPWELLING SYSTEM" by Michelle Graco, Sara Purca, Boris Dewitte, Carmen G. Castro, Octavio Morón, Georgina Flores, Jesús Ledesma and Dimitri Gutiérrez. We have revised the manuscript following the reviewers' suggestions. All recommendations have been taken into account. In particular, we have expanded the analysis to include the 2010-11 La Niña event so that the data are now covering the period 1996-2011. Some sections have been completely rewritten and we have thoroughly checked over the grammar and syntax, and improved the English so that the paper should be now friendlier to read.

Note that our manuscript is aimed at contributing to the Special Issue: Biogeochemical processes, tropospheric chemistry and interactions across the ocean/atmosphere interface in the coastal upwelling off Peru.

We hope that you will find our revised manuscript now suitable for a publication in Biogeosciences.

Best regards

Michelle I. Graco

Michelle Graco

**Reply to reviewer 1**

*As mentioned above, the time series are great. In the text it is stated, that the records are used until the end of 2009. No information is given, that the measurement program stopped. In case the measurements continued until the end of 2011 it would improve the manuscript considerably to extend the time series.*

We have expanded the analysis to include the 2010-11 La Niña event so that the data are now covering the period 1996-2011.

To extend the IEKW for 2011, we used the outputs of another general circulation model simulation than SODA. This simulation belongs to the set of MERCATOR global OGCM simulations (http://www.mercator-ocean.fr/) and was used recently in Mosquera et al. (2014) to analyze the equatorial Kelvin wave in the Pacific.

Following the reviewer's recommendation, we have rewritten the description of the method to derive the Kelvin wave, which should be clearer now.

References added to the revised manuscript:

Illig S., B. Dewitte, K. Goubanova, G. Cambon, J. Boucharel, F. Monetti, C. Romero, S. Purca and R. Flores, 2014: Forcing mechanisms of intraseasonal SST variability off Peru in 2000-2008: local versus remote forcings. J. Geophys. Res.-Oceans, Vol. 119, 6, 3548-3573.

Mosquera-Vásquez, B. Dewitte and Serena Illig. The central Pacific El Niño intraseasonal kelvin wave JGR: Oceans (2014). 10.1002/2014JC10044.

*It is mentioned that in 2002 the measurements changed from Niskin bottles to CTD measurements, hence a discussion how this change in method influences the measurements is needed.*

The reviewer is right; the change in the technique could be influential on the data set. In fact, after 2002 we kept measuring temperature and salinity using both the CTD and the Niskin bottles to verify that there is no off-set from before and after 2002. In general, both measurements are very close.

*Normally an abbreviation is defined and then used in the following text. In this manuscript, the repeated definitions of the same abbreviations are strange and make the manuscript difficult to read. E.g., IEKW is defined on page 7 two times in lines 1 and 2, on page 8 line*

18, page 11 line 27 and page 17 line 1. ESSW is defined page 3 line 8, page 7 line 25 and page 9 line 10. Similar for other abbreviations.

It was corrected accordingly.

The units used in the figures and text should be the same. Oxygen in the text is presented as micromole/kg (e.g. page 9 line 8), in figure 3 as micromole/L; for nutrients in micromole/L (e.g. page 10, line 14) in Figure 5 as microM.

It was corrected accordingly.

May be use months instead of seasons to better define the period you refer to.

It was corrected accordingly.

Page 10 line 28 to page 11 line 2: The changes mentioned are difficult to see in Figure 5d, may be add the depths range in the text you are considering.

It was corrected accordingly.

Page 10 lines 3 to 5. PC1 And PC2 are mentioned and shown in Figure 6b. In paragraph 2.3 (page 6) where the method is described, only PC1 is mentioned. PC2 should be also explained in paragraph 2.3.

It was corrected accordingly.

All the specific comments and minor changes in the text and figures were taken into consideration in order to improve each section.

**Reply to reviewer 2**

The paper addresses an interesting topic of physical-chemical coupling in the northern Humboldt System. This region is particular attractive because of its strong connection with equatorial dynamics, high biological productivity, as well as a shallow and intense oxygen minimum zone that influences nitrogen cycling and biological dynamics. The study examines

the temporal and vertical variability of relevant physical and chemical variables during the period 1996-2009, covering the strong El Nino 1997-98 and subsequent La Nina 1998-99, as well as period with relatively neutral or weak ENSO conditions after 2002. The authors conclude that most of the physical-chemical variability off Peru is linked to equatorially originating remote forcing. They also suggest enhanced physical-chemical intraseasonal variability after 2002, associated with a change in the intraseasonal equatorial Kelvin wave (IEKW) activity. None of those statements is well supported in the paper. Three aspects could be argued against the conclusions.

We thank the reviewer for his constructive comments. We have revised the manuscript to clarify these points. In particular, we have quantify this through the EOF combined analysis done over the full period (including the 1997/98 El Nino event) and an additional EOF analysis for a reduced period that exclude the 1997/98 El Nino event) (i.e. 2000-2011). The nature of the coupling (as inferred from the eigenvalues of the EOF model) is also significantly impacted (See new Table 1). We have also included a new Table 2 that presents the correlation between the PC timeseries of the EOF analysis and the indices of El Niño and the equatorial Kelvin wave supporting our interpretation.

First, since the local forcing (e.g. coastal winds, heat fluxes) was not examined; it is not possible to conclude that remote forcing is the dominant forcing.

We agree with the reviewer that a limitation of our study is to disregard the role of local wind forcing. Following the reviewer's suggestion, we have expanded the discussion on the role of wind forcing based on previous studies (Dewitte et al 2011, Echevin et al 2014; Illig et al., 2014) (see discussion section). In our study we are not addressing the higher-frequency variability associated to wind forcing dueto the lack of *in situ* data over this period and limitations of the satellite products (see introduction of Goubanova et al. (2011) for a discussion on this topic). We have toned down some of the statement regarding this mater, but, we maintain that the variability of our data set can be to a large extent understood in terms of the oceanic teleconnection from the equatorial region, which is consistent with recent modeling studies (Dewitte et al., 2012; Illig et al., 2014; Echevin et al., 2014) and former observational works (Gutierrez et al., 2008).

References:

Dewitte B., J. Vazquez-Cuervo, K. Goubanova, S. Illig, K. Takahashi, G. Cambon, S. Purca, D. Correa, D. Gutiérrez, A. Sifeddine, and Ortlieb, L.: Change in El Niño flavours over 1958-2008: Implications for the long-term trend of the upwelling off Peru. Deep Sea Research II, 123-135, 2012.

Echevin, V., Albert, A., Lévy, M., Graco, M., Aumont, O., Piétri, A. and Garric, G.: Intraseasonal variability of near shore productivity in the Northern Humboldt Current System: The role of coastal trapped waves. Continental Shelf Res.,7314–30, 2014.

Gutiérrez, D., Enríquez, E., Purca, S., Quipuzcóa, L., Marquina, R., Flores G., and Graco, M.: Oxygenation episodes on the continental shelf of central Peru: remote forcing and benthic ecosystem response. Prog. Oceanogr. 79, 177–189, 2008.

Illig S., B. Dewitte, K. Goubanova, G. Cambon, J. Boucharel, F. Monetti, C. Romero, S. Purca and R. Flores,: Forcing mechanisms of intraseasonal SST variability off Peru in 2000-2008: local versus remote forcings. J. Geophys. Res.-Oceans, Vol. 119, 6, 3548-3573, 2014.

Second, the coastal time series have monthly resolution that precludes a proper characterization of the dominant 40-60 day intraseasonal variability off Peru, making difficult to connect the physical-chemical changes off Callao to the IEKW variability.

We agree that the monthly resolution will tend to damp the amplitude of the intraseasonal variability. However since we focus on the frequency band [40-90] days, we believe that it is still reasonable to address the intraseasonal variability with the monthly data. The potential limitation associated to aliasing that would require to be investigated further based on mooring data. There is however no biogeochemical data from a long-term mooring available in this region. The establishment of a long-term mooring is one of the future actions planned by IMARPE with the international collaboration.

Third, the characterization is strongly dependent on El Nino 1997-98 and subsequent La Nina disturbances. It is not clear for me if a novel scenario exists after 2002.

We have completely re-written the discussion section. The time series presented here illustrate two different regimes for the OMZ dynamics and biogeochemical activity off Peru. One regime with a strong asymmetry associated to the extreme Eastern Pacific 1997- 1998 El Niño when the OMZ disappeared in the upper layer and the subsequent intensification of the OMZ and nitrogen lost during 1999-2001 La Niña. Other regime since 2000 characterized by a strong intraseasonal variability in the intensity of the OMZ and the availability of nutrients, loss nitrogen processes; and a tendency for a decoupling between the chemistry and the physical forcing associated with weaker but more frequent warm events. We hope that we have stated clearly in this new version of the manuscript.

The introduction is weak. I suggest be more concise and define better the paper goals. The sentence "in order to infer potential biogeochemical scenarios in connection with equatorial variability" appears to me too vague.

Following the reviewer's suggestion, we have presented a more concise introduction and have clarified the objectives.

The instruments to measure temperature and salinity changed in 2002. Could that explain the apparent change in the physical-chemical coupling after 2002? Do you have a period with overlapping observations of thermometer, salinometer, and CTD to check the measurements consistency?

After 2002, we kept measuring temperature and salinity using both the CTD and the Niskin bottles to verify that there is no offset from before and after 2002. In general, both measurements are very close.

Is the EOF analysis really needed? Only the PC time series were shown, what about the spatial pattern?

The EOF analysis is performed in order to summarize the variability for physical and biogeochemical parameters. The results on the EOF analysis have been completely rewritten and should be clearer in this new version. We have also expanded this analysis to support the idea of the two regimes. Following the reviewer's suggestion, we also provide the eigenvalues of the EOF modes (spatial patterns) which are informative on the way the variables relate to each other (See Table 2).

Results. Page 8 Line 17: Year 2008 is not El Nino according to El Nino 3.4 index. Lines 22-24: How do you support the Z_15_C - ENSO connection?

We agree with reviewer's statement that the El Niño 3.4 index in 2008 does not indicate a warm event. However, the costal El Niño index 1+2 indicates a warm period, which is related to the crossing of downwelling Kelvin wave. The Peruvian coast is one of the few regions in the world that requires two indexes for monitoring El Niño. In 2012, the national technical committee for El Niño study (ENFEN; http://www.met.igp.gob.pe/variabclim/enfen/) defined the ICEN ( Coastal El Niño index) based on the anomaly of the sea surface temperature of the El Niño region 1+2 (90° W- 80°W, 10°S-0°). The ICEN index is better indicator of the ENSO cycle off the Peruvian coast. It gives an idea not only of the El Niño impact on the physical and chemical fields but also on the El Niño consequences on the biota and consequently on the economic resources. We will include this in the methodology.

We have also extended the discussion indicating that after the 1997 - 98 El Niño, the interannual variability in the equatorial Pacific consists in a different type of El Niño events. Those events are referred to as Central Pacific El Niño events and are characterized by an

increased variability of the intraseasonal Kelvin wave activity during the development and peak phase compared to the strong El Niño events (See Mosquera et al. (2014). Therefore, the period 2000-2011 can be considered as a period with enhanced intraseasonal variability compared to the previous decade.

It possible to include numerical model outputs in the analysis? Several regional model efforts have been done in the region that might be useful to interpret better the observations.

We agree that model experiments could provide additional material for interpreting the observations. For OMZ regional modeling, we are only aware of the study by Montes et al. (2014) and Vergara et al. (2016) that addresses the seasonal cycle, not the interannual variability. It is beyond the scope of the present study to present model simulations over the period of interest, which would deserve a thorough validation. This is actually a work in progress and the results will be reported shortly.

I do not understand why do you use only the global wavelet spectrum. Examining wavelets in the time-frequency domain is more informative, especially to see temporal changes in the IEKW activity.

We perform an analyze in the time frequency wavelet frequency in addition to the global wavelet spectrum and finally we decide to use the GWS because both convey the same information, that is  the strong interannual signature associated with El Niño and  the intraseasonal signature that appear in the band of 180 days (6 months)-90 days (3 months). In the Figure 8, GWS showed the significance interannual periods for all-time series, over the dotted line.

Overall the discussion needs a better integration of biogeochemical processes influencing the observed vertical profiles of oxygen and nutrients.

Following the reviewer's suggestion, we have improved the discussion on this matter.

All the specific comments and minor changes in the text and figures were taken into consideration in order to improve each section.

---

## Author Response (AR2)

Lima, the 23 April, 2017

Michelle Graco
Department of Oceanography
Instituto del Mar del Perú (Imarpe)
Esq. Gamarra y Gral Valle s/n
Chucuito-Callao, Peru
Phone/Fax: 0051-1-429 6069
e-mail: mgraco@imarpe.gob.pe

Dear Dr Herndl,

We highly appreciated all the comments and suggestions from both reviewers about our manuscript entitled "THE OMZ AND NUTRIENT FEATURES AS A SIGNATURE OF INTER-ANNUAL AND LOW-FREQUENCY VARIABILITY OFF THE PERUVIAN UPWELLING SYSTEM" by M. Graco and co-authors. We have revised the manuscript following all the reviewers' suggestions and recommendations. In particular, we now focus on the inter-annual variability and have left behind the results on the intra-seasonal variability considering that the data have limitations that are now discussed more thoroughly. New diagnostics are provided to expand the section on the role on ENSO and the identification of the two regimes.

Some sections have thus been completely rewritten and new figures were included. We have thoroughly checked over the grammar and syntax, and improved the English so that the paper should be now friendlier to read.

We hope that you will find our revised manuscript now suitable for a publication in Biogeosciences.

We are grateful to the editorial office for his understanding in the review process and the works of the reviewers.

Best regards

Michelle I. Graco

**Reply to reviewer 1 (Report 2)**

**General comments:**

The revised manuscript by Graco et al. has improved compared to the first version. However, there are still some shortcomings, which need to be solved before the manuscript can be published. The connection of the measurements to El Nino and La Nina periods is weak and should be strengthened (see below). Very disappointing is the fact that none of the 7 !!! co-authors seemed to have looked carefully at the revised manuscript (see below) and left the work for the reviewers as was the case already at the first version. The English is still bad (I know all authors are no native English speakers, but some help should be searched to make the manuscript better readable). I recommend revision of the manuscript.

We thank the reviewer for his constructive comments. We have improved the manuscript following the recommendations. In particular we have strengthened the section on the link between $O_2$ and ENSO/Kelvin wave activity (see also the specific comments). Note that we have also expanded the section presenting the data by now showing anomalies relative to the mean seasonal cycle and the mean seasonal cycle for T and $O_2$ (new figure 3), which serves as material for introducing the analysis on the relationship between OMZ variability and the equatorial Kelvin wave and ENSO.

The text has also been modified in many parts and proofread in order to improve the spelling and make it easier to read.

**Specific comments:**

So far, the authors missed the opportunity to well relate the measurements to the El Nino and La Nina periods. In line 242 a 1997-1998 El Nino event and a 1999-2001 La Nina event are mentioned but these events are not clearly defined and other weaker events are not really referenced. According to the ONI index El Nino's during the time period presented are: 5/97-5/98 (defined as strong), 6/2002-2/2003 (moderate), 7/04-4/05 (weak), 9/06-1/07 (weak), 7/09-4/10 (moderate) and La Ninas 7/98-3/01 (moderate), 8/07-6/08 (moderate), 7/10-4/11 (moderate) and 8/11-3/12 (weak). These events are well visible in the parameter distribution, especially in the N deficit and the 15°C time series. Hence the El Nino and La Nina time periods should be mentioned in the text and related to the figures instead of listing some years with deepening of the thermocline and mention a coincidence with El Nino conditions (lines 233 to 237). In line 302 the end of 2011 was described as strong La Nina condition, but according to the ONI index it was only a weak La Nina.

Following the reviewer's recommendation, we have clarified our methodology. In particular we provide a table (new table 1) summarizing the characteristics of the inter-annual events over the period of interest (moderate versus strong, Central Pacific El Niño versus Eastern Pacific El Niño). The selection of the El Niño years and El Niño types follows Yu and Kim (2013).

**Table 1**: El Niño and La Niña years and types of the event

| El Niño (red) and La Niña (blue) years | Magnitude (Yu and Kim, 2013) | Type (Eastern Pacific versus Central Pacific) |
|---|---|---|
| 1997/1998 | Strong (extraordinary) | EP |
| 1998/1999 | moderate | |
| 1999/2000 | moderate | |
| 2000/2001 | weak | |
| 2002/2003 | moderate | CP |
| 2004/2005 | moderate | CP |
| 2006/2007 | moderate | Mixed |
| 2007/2008 | moderate | |
| 2009/2010 | moderate | CP |
| 2010/2011 | moderate | |

References:

Yu J.-Y. and S. T. Kim, 2013: Identifying the types of major El Niño events since 1870. Int. J. Climatol., 33:2105–2112. doi:10.1002/joc.3575.

**Here is a list of errors that a co-author should have noticed in case of reading the manuscript:**

We thank the reviewer for his thorough review. We have proofread the manuscript and we hope it is now friendlier to read.

-line 250 Figure 3a. This is still a reference from the first version of the manuscript, now it is Fig. 2c. Similar on line 297 it has to be figure 2 not figure 3.

It was corrected

- Line 316: 'see horizontal lines in figure 6a,b'. In the revised version there are no horizontal lines marked for the standard deviation as had been in the first version of the manuscript.

It was corrected.

-Line 345: 'Z_ZMO (Fig. 6 d)'. Figure 6d is Z_OMZ.

It was corrected.

- Line 359: 'Figure 7b'. In Figure 7 no a,b,c, are marked.

It was corrected.

- Line 438: In the reply-letter it is stated that the units will be used always the same, but in line 438 and 796 the unit for oxygen are different compared to the text.

It was corrected.

- Figure 6 legend is not correct, (b) is now (c); (c) is now (d), (d) is now (e). Figure legend should be adjusted when the figures modified.

It was corrected.

- Figure 7 legend: As mentioned above, no a, b, c is marked in the figure but mentioned in the legend.

It was corrected.

- The author wrote in the reply-letter that repetition of definitions were removed. However, ESSW (line 220) was already defined in line 63, SSW (line 254 and line 426) already in line 227, IEKW_1 and IEKW_2 (line 311) already in lines 189/190, while CTW (line 521) is not defined.

It was corrected.

- References were not checked: Line 464, there is no Naqvi 1991 in the reference list (only Naqvi and Noronha 1991) line 465 no Codispoti et al 1985 (only Codispoti and Christensen 1985), line 468 no Morales et al 1996. Kalvelage et al. 2013 in the reference list is not cited in the text.

It was corrected.

-In the abstract (line 30) acidic oxygen minimum Zone is mentioned. In the entire text acidic is not mentioned again. Even if not investigated in this manuscript it should be briefly mentioned in the longish general review.

The world acidic was removed.

-May be define SST in line 89.

It was corrected.

-Line 123 '20 nm' is strange; better write '20 nm off the coast'.

It was corrected.

-Line 140: I don't understand the meaning of 'of the Oxygen Minimum Zone- OMZ and changes…'

The paragraph was removed.

-Line 221 'during summer and spring periods', mention austral summer and austral spring as you do later in the text.

It was corrected.

-Line 227-228: SSW can't 'penetrate into the coast', may be write 'penetrates into the near-coastal area' or whatever you like to describe.

It was corrected.

-Line 258: Better include 'depth' after 'thermocline'.

It was corrected.

-Define 'N.S' in line 262 and in the header of table 2.

It was corrected.

-Line 320: 'delayed by 1 month', Where can I see this?

This statement is based on the consideration that the Kelvin waves of the first and second baroclinic modes are both forced in the central-western Pacific but propagate at different speeds (~2.8 m/s for mode 1 and ~1.7 for mode 2). Assuming that they propagate freely from the central-western Pacific, the difference in phase speed yields a ~1 month delay between mode 1 and mode 2 when they reach 95°W. We provide the figure that shows the mode 1 and mode 2 Kelvin Waves at 95°W. The maximum correlation is for lag -1 month (mode 1 ahead mode 2).

[Figure]

 As I am not able to access all the grey literature listed in this manuscript, is '3°N' correct, as the title states costa peruana, and the Peruvian coast starts south of

The reviewer is right. The northern border of Peru is at ~3°S. However, in the title of this publication, the authors refers to 3°N because the cruise was done at 3°N.

-Line 775: Define E and C in the table header and not only in the text.

Following the reviewer recommendation, we now also define the E and C indices in the caption of the Table.

-Figure 1 legend: Mention station 4 as 12°02'S; 77°29'W as in the text, not only by one geographical coordinate.

It was corrected.

-Figure 2 legend: Mention 15°C marked in Figure 2a to indicate the lower thermocline limit. In the figure the numbers are difficult to read.

The figure 2 was improved.

-Figure 3a: The numbers are difficult to read. May be describe in the figure legend the curves, e.g. 15°C (solid black line), etc.

It was corrected.

-In Figure 4 legend it is stated that that the oxylines 22.5 and 45 are included in the figure; I can't find them in the figure.

The caption of Figure 4 was corrected.

-Why is there a black line in Figure 5 at 60 m depth?

The black line was removed. It was an error.

**Reply to reviewer 2 (Report 1)**

The paper improved significantly from its first version. Objectives, structure, figure layout, and narrative are much clearer now. Still, English grammar/usage and several confusing paragraphs need revision. Overall, I think that the paper is valuable for publication after some clarifications. I strongly suggest an extra analysis exploring the link between local wind and the biogeochemical time series variability. Besides, I am not convinced that a monthly time series can resolve the whole intra-seasonal band (30-90 days). I have attached a list of issues that need clarification or a better justification.

We thank the reviewer for his constructive comments. We have proofread the manuscript so as to make it easier to read. We have also clarified and expanded some sections. In particular the presentation of the data for T, S and $O_2$ has been expanded so as to better introduce the subsequent analyses and ease the interpretation of the inter-annual variability. In particular, we now provide the new figure 3 that shows the evolution of the anomalies of T and $O_2$ relative to the mean seasonal cycle, and of the seasonal cycle as a function of depth, along with statistics.

Regarding the issue of the intra-seasonal band, we have also considerably revised the manuscript and acknowledged the fact that it is indeed difficult to quantitatively address variability at the intra-seasonal time scales with our oceanographic data. The paper thus now focuses on the inter-annual timescales. Still, we believe that intra-seasonal variability not accounted by our data can have a residual effect on the lower frequency signal (i.e. inter-annual) through rectification processes, which we now briefly discuss in the discussion section.

Since the focus is now on inter-annual timescales, we do not investigate the relationship with local winds owned to the fact that the oceanic teleconnection is notoriously dominant at inter-annual timescales and limitations associated to wind products over the period of interest (see Goubanova et al. (2011) for a discussion).

References:

Goubanova K., V. Echevin, B. Dewitte, F. Codron, K. Takahashi, P. Terray, M. Vrac, 2011: Statistical downscaling of sea-surface wind over the Peru-Chile upwelling region: diagnosing the impact of climate change from the IPSL-CM4 model. *Clim. Dyn.,* DOI 10.1007/s00382-010-0824-0.

**Specific comments**

-Multiples times the authors used "owned" to indicate "due" or "as a consequence" (or any proper synonym related to the last two verbs). Need correction.

This has been corrected accordingly.

-Introduction

-70-72: Could you explain why oxygen determines low N:P ratios?

Under the low oxygen conditions of the coastal upwelling waters, other redox processes can be dominant such as denitrification and anammox. Both processes makes use of nitrate as an electron acceptor that determines the high rates of nitrogen transformation in $N_2$ that is exchanged with the atmosphere. This results in alow N/P ratio, below the classical Redfield ratio of 16.

-83-84: What type of reduction are you talking about?

Sanchez et al. (1999) reported that during the 1997– 1998 El Niño event large scale oxygenation off the Peru margin was caused by circulation changes. The latter led to the depression of the OMZ below 100 m. Under these circumstances, the OMZ off Peru and northern Chile could be reduced by 61% (Helly and Levin, 2004). We have re-written this sentence in the new version.

-Results

-Figure 3:

a) The three series together are hard to compare. I suggest either using more contrasting color for the time series, or showing only the thermocline and oxycline depths (or thermocline and OMZ depths, opt to you).

Figure 3 has been changed so as to clarify this point and also to ease the interpretation of the variability. We now present the anomalies of (T,$O_2$) as a function of depth (with the thermocline and oxycline depths overlapped) along with the mean seasonal cycle as a function of depth. Mean seasonal cycles have been calculated over the period 1999-2011 so as to exclude any residual effect of the strong 1997-1998 El Niño. A new figure 3 is shown.

b-d) I suggest adding the 1st and 3th quartiles, so we can have an idea of the data dispersion for each month.

We have modified figure 3 and the seasonal cycle is now presented as a function of depth. The seasonal cycle is computed over the period 1999-2011 to avoid the strong signature of the 1997-1998 El Niño and there is little dispersion associated to its estimate. For instance, if we select randomly 10 years out of the 13 years and recalculate the seasonal cycle, we find little differences.

-246: Why I should expect a shallow oxycline condition for summer-fall. Need to explain.

In summer-fall, the shallow oxycline is due to the high remineralization of organic matter, higher stratification and oxygen consumption. Note that oxygen poor waters can intercept the euphotic layer and the continental shelf, promoting suboxic and even anoxic conditions in the underlying surface sediments ($O_2$ < 8.9 μmol kg$^{-1}$, Fig. 2c). See also Vergara et al. (2016) for a description of the seasonal variability off Callao from a model simulation (their figure 13).

-250: "deficient water deepens" suggests vertical advection, but that is not necessarily the case. Subsurface oxygen can increase due to vertical mixing. Also, decreased phytoplankton production in winter leads to decreased subsurface respiration, so lower oxygen consumption.

During austral winter (July-August) due to higher variable winds (See Fig 1b of Dewitte et al. (2011)), there is an increase in mixing that is associated to a lower phytoplankton production, which yields to the oxygen deficient waters (up to 70 m depth, figure 2c and 3).

-256-257: Defining intra-seasonal as fluctuations shorter than a year is not correct. That would include the seasonal cycle, which is evidently not intra-seasonal.

The text has been clarified. Intra-seasonal timescales refer to timescales in the frequency band [30-120] days.

-263: "Decoupling in the linear sense" Need to explain.

"Linear sense" refers to the correlation analysis that provide the estimate of the linear relationship between two variables. The fact that the correlation is low means that there is no linear relationship. However since we are dealing with a highly non-linear system, this does not mean that there is not a causal relationship between the oxycline and the 15°C isotherm.

-264: Are those trends statistically significant? Two cm per decade is really small, especially considering potential measurement errors. It may be interesting to calculate the linear trend independently for each month or season.

Following the reviewer suggestions, we have estimated the significance of the trend and its seasonal variability (see results). The new table 3 was introduced in the revised manuscript:

Table 2: slope of the linear fit for oxygen, temperature,salinity, nitrate and nitrite as a function of depth over the period 1999-2011. The slope for thermocline and oxycline depths are also provided as a function of season. The confidence level estimated based on a Student's T-test is indicated in parenthesis when larger than 80%.

| Depth (meter) | O$_2$ (µmol kg$^{-1}$ decade$^{-1}$) | T (°C decade$^{-1}$) | S (PSU decade$^{-1}$) | Nitrate (µmol L$^{-1}$ decade$^{-1}$) | Nitrite (µmol L$^{-1}$ decade$^{-1}$) |
|---|---|---|---|---|---|
| 0 | 24.03 (90%) | -0.04 | 0.026 | 0.93 | 0.11 |
| 10 | 47.55 (95%) | 0.53 (80%) | 0.013 | -0.17 | -0.22 (80%) |
| 25 | 40.35 (95%) | 0.65 (90%) | 0.025 (80%) | -1.67 | -0.01 |
| 50 | 14.40 (85%) | 0.50 (90%) | 0.003 | 0.01 | -0.15 |
| 75 | 6.04 (90%) | 0.34 (90%) | -0.002 | 1.85 | -0.57 |
| 90 | 6.76 (95%) | 0.42 (95%) | -0.001 | 2.51 (80%) | -0.75 |
| 100 | 7.53 (95%) | 0.46 (95%) | 0.003 | 2.95 (80%) | -0.88 |

| Annual | OMZ (m decade$^{-1}$) | | Thermocline (m decade$^{-1}$) | |
|---|---|---|---|---|
| | -0.64 (95%) | | -0.30 (95%) | |
| Seasonal | Summer | Winter | Summer | Winter |
| | -0.74 (95%) | -0.77 (95%) | -0.63 (95%) | 0.03 |
| | Fall | Spring | Fall | Spring |
| | -0.76 (95%) | -0.69 (95%) | -0.49 (95%) | -0.48 (95%) |

-272-274: Could you indicate which depth intervals correspond to surface, subsurface, and bottom waters? That could help to better understand the time series description.

Following reviewer´s recommendation, the caption of Figure 2 was expanded so as to provide more details on the data.

 -276: I would mention strongest vertical mixing instead of stronger upwelling. That is finally the most likely mechanism explaining the pattern.

We are referring here to the distribution of Nitrate into the water column, so that the most likely mechanism is upwelling. During winter, high nitrate concentrations are found over the entire water column (> 15 µmol L$^{-1}$). Although the Peruvian upwelling benefit from upwelling favorable winds during all the year, it is during the winter season when winds are the strongest leading to maximum upwelling and high variability which implies the mixing. Both processes are likely to be at work to bring nitrate to the surface.

-285: Explain why phosphate is not a limiting nutrient.

Nitrogen appear more limiting than phosphate because an active N loss is observed in large parts of the water column promoted under low/depleted oxygen conditions and coincident with a pronounced secondary nitrite maximum and a strong nitrogen deficit (Codispoti et al., 1986).

-286-287: "whereas nitrate is depleted at surface and in the water column" Confusing statement, describe better. Why do you cite Kock et al. (2016) to describe the nitrate depletion in your dataset?

Following reviewer´s recommendation we have clarified this statement and write a new paragraph.

-288: "registered the highest nitrate values" delete nitrate

It was corrected.

-289-290: First you talk about nitrate concentration, and then nitrite variability. You cannot indicate 'also' to connect the two sentences. Besides, I do not see an increase in nitrite variability after 2000.

The text has been clarified. "After 2000, nitrate concentrations registered the highest values (> 20 µmol L$^{-1}$) for the entire time series coincident with lower nitrite and silicate concentrations (Figure 4)."

-292-299: Ndef was estimated using a constant of 12.6. Is that constant derived for the Peruvian coastal region only? Does it change in the oceanic region? If that is the case, could you conclude that there was low nitrate consumption during El Nino 1997-98, when oceanic Subtropical Surface Waters penetrated into the coastal region?

The constant of 12.6 was defined for the Peruvian waters, particularly in the coastal areas. This value is lower than in oceanic waters (Codispoti and Packard, 1980). There are a lot of grey literature that propose this non-Redfield relationship in the Peruvian waters. This condition is the consequence of the low oxygen (OMZ) and the high nitrogen loss under denitrification-anammox activity.

During El Niño there are Subtropical Surface Waters (SSW) coming to the coast. Low nutrient conditions are characteristics of these waters compared with the Cold Coastal Waters (CCW), and high-oxygen prevents the nitrogen loss and high nitrogen deficit.

-299: "Nitrate reduction" is confusing. Do you mean that nitrate concentration decrease, or are you talking about a redox reaction?

The depleted oxygen conditions lead to an increase of the redox. Nitrate is one of the electron acceptor available in these waters.

-303: isotherm depth instead of isotherm

It was corrected.

-312: "These waves differ from their vertical…" sentence does not make sense

The sentence was rewritten: "These waves have a different vertical structure and phase speed"

-319: Could you explain why a downwelling coastal-trapped wave decreases wind-driven upwelling? A CTW can modify the source of water being upwelled, but the wind driven transport remains the same if wind does not change.

What is meant here is that a downwelling coastal-trapped Kelvin wave will depress the thermocline depth along the coast, which opposes to the effect of the upwelling favorable winds. The text has been clarified.

-333: What do you define by low frequency variability?

This statement was removed since we no longer discuss intra-seasonal timescales. Note however that the EKW_2 tends to have more energy in the low periods (~4-8 years) than EKW_1 (see figure 7)

-338: I do not understand when you say "skewed from 2000". Why is noteworthy?

The skewness refers to the asymmetry of the timeseries. A negative skewness means that the amplitude of the negative anomalies is larger in absolute value than the amplitude of the positive anomalies, or that there are more occurrence of negative anomalies than positive anomalies. This is now quantified and we provide the value of the skewness:

"It is noteworthy that EKW_2 is negatively skewed from 2000 (normalized skewness = -0.8910 cm) and there is a negative trend of upwelling events from 2000 (trend = -0.0177 cm/decade)"

-344-345: Confusing. I could figure out what do you mean after examining Figure 6, but the text is not clear at all.

The text has been clarified.

-351: It is not possible to resolve fluctuations shorter than 60 days with your monthly observations (1/60 day-1 is the Nyquist frequency), so why do you talk about the 50-180 days band. I do not even think that you could properly resolve fluctuations shorter than 90 days.

The text has been corrected accordingly. Intra-seasonal variability refers to timescales between 60 days and 180 days. We no longer discuss intra-seasonal variations in the data considering their limitations (gaps and aliasing). This is now explicitly mentioned in the text of the revised manuscript.

-Figure 6.

What are the y-axis units in panels f-j. I cannot understand the result if I do not know what the y-axis represents. Delete c) in the titles from panels 6f-j.

The figure was improved.

-362: Why is this interesting? What does it imply?

This section has been completely rewritten and the text has been clarified.

-365: I would rather indicate that the EOF2 has weaker low-frequency variability compared to EOF1, instead of stronger high frequency variability the text was modified accordingly.

-369: Where that 'also' comes from? It does not make sense, since you indicated a different feature for the EOF1 in previous paragraph

We apologize for the misuse of "also" here. We meant "in addition".

-371: Why is this interesting? Explain.

This section has been completely rewritten and the text has been clarified.

-380: How do you visualize that the mode tends to capture higher frequencies?

This section was rewritten and we no longer discuss intra-seasonal variability

-384-385: "over the period after 2000" do you mean "after 2000"?

Yes. This has been changed accordingly.

-388: The main conclusion is not clear. What are those different regimes? How do you connect your results with the "two regimes" statement?

This part was presented in the discussion and clarified.

**-Discussion**

395: "with in particular a marked semi-annual cycle" does not make sense. Do you mean "particularly, with a marked semi-annual cycle". I disagree that the OMZ has a strong semi-annual cycle. The semiannual fluctuation exists but is weak. Please show the interquartile range or the standard deviation in Fig. 3b-d.

We now provide a new figure 3 that shows the evolution of the anomalies of T and $O_2$ relative to the mean seasonal cycle, and of the seasonal cycle as a function of depth, along with statistics.

-399: delete "Firstly,"

done

-400: You did not explore the connection between the equatorial variability and the OMZ in the seasonal time scale. This is kind of surprising considering that your biogeochemical time series can resolve properly the seasonal band (which is not the case for the intra-seasonal). That the 15°C isotherm depth in the equator has similar semiannual cycle that the OMZ off Peru does not mean that a connection exists.

The paper focuses on the ENSO timescales and we have left behind a thorough discussion of the seasonal cycle because the latter is not dominated by the oceanic teleconnection and would require additional analyses that are beyond the scope of the current study.

-406-408: Could nitrification contribute to the subsurface peak in nitrite?

Yes this is possible. Molina and Farías (2009) indicate that aerobic NH4+ oxidation could contribute between 8% and 76% of NO2− production and support the important role of aerobic NH4+ oxidizers in the nitrogen cycling in the OMZ and at its upper boundary.

**2. Methods**

**2.1. Time series**

A monthly time series (1996-2011) of vertical profiles of temperature, salinity, oxygen and nutrients (nitrate, nitrite, phosphate and silicate) collected by the Instituto del Mar del Peru (IMARPE) during the period 1996-2011, at station 4 (20 nm, 145 m), located off central Callao (12°02' S, 77°29´ W, Figure 1) was used. Filtering of the data was performed by eliminating statistical outliers. The time series were standardized with their annual cycle in order to explore several temporal scales of variability in the study area.

The region of Callao have been identified as one of the major upwelling cells off central

Peru (Rojas de Mendiola, 1981) with a subsurface well developed OMZ (Wooster and

Gilmartin, 1961; Zuta and Guillén, 1970). The presence of nitrate-rich and oxygen deficient

ESSW (Zuta and Guillén, 1970; Strub et al., 1998; Graco et al., 2007; Silva et al., 2009) triggers the high primary production of the region, with maximum values in spring-summer, out of phase of winter upwelling maximum (Echevin et al., 2008; Chavez and Messié, 2009; Gutiérrez et al., 2011a, Vergara et al., 2016). The OMZs are associated with areas of intense microbial activity associated with organic matter remineralization, oxygen consumption and nitrogen loss (Helly and Levin, 2004). Recent studies in the Peruvian oxygen minimum zone show not only denitrification but also the anammox process as the prevalent pathway for nitrite and nitrate reduction and a loss of nutrient available for phytoplankton. Others studies have documented the impact in the oxygen regime on the continental shelf off Callao of the remotely-driven effects of El Niño and coastal trapped waves that show effects at monthly to interannual time- scales on the boundaries of the Oxygen Minimum Zone- OMZ  and changes in the benthic fauna (Gutiérrez et al., 2008).

**2.2. Water column profiles**

Water samples were collected monthly with Niskin bottles during cruises of R/V

IMARPE VIII, R/V SNP-1 and R/V SNP-2 for the period 1996 to 2011. Temperature was measured by inversion thermometer through 2001 and by CTD (Seabird SBE 19+) from 2002.

Salinity was measured by salinometer through 2001 and by CTD plus salinometer from 2002.

IMARPE performs comparative analysis between CTD, thermometer and salinometer during all the monthly cruises in order to present consistent information.

Dissolved oxygen and nutrients were measured at standard depths (0, 10, 30, 50, 75,

100 m). Dissolved oxygen was determined by a modified Winkler method (Grasshoff et al.,

1999), with a precision of 0.5 $\mu$mol kg$^{-1}$ (large errors for values lower than 10 $\mu$mol kg$^{-1}$).

Nutrient samples (nitrate, nitrite, phosphate and silicate) were stored frozen until analyses using standard colorimetric techniques, the precision for nitrate analysis was $\pm$0.5 $\mu$mol L$^{-1}$, for nitrite

$\pm$0.08 $\mu$mol L$^{-1}$, for phosphate $\pm$ 0.03 $\mu$mol L$^{-1}$ and $\pm$0.25 $\mu$mol L$^{-1}$ for silicate (Parsons et al.,

1984).

The fixed nitrogen deficit (Ndef) was determined by the formula:

Ndef= 12.6 x [ HPO$_4^{2-}$] – ([NO$_3^-$]+[NO$_2^-$])

The constant 12.6 is the empirically-determined N:P ratio of organic matter produced in these waters (Codispoti and Packard, 1980). Positive values indicate nitrate deficit.

The OMZ was defined as the area with oxygen concentrations lower than 22.5 $\mu$mol kg$^-$

$^1$. This concentration was considered as the OMZ upper boundary (Schneider et al., 2006;

Fuenzalida et al., 2009; Ulloa and Pantoja, 2009).

**2.3. Statistical analysis of Time series**

Linear interpolation was applied to complete minor gaps into the 1996-2011 time series.

Monthly anomalies were calculated as the difference between the original data and the mean seasonal cycle over the full period, normalized by the standard deviation. To compare variances, the time-series were analyzed with discrete wavelets (Torrence and Compo, 1998) to represent temporal changes in spectra. The principal component of the first two dominant Empirical

Orthogonal Function (EOF) analysis (Emery and Thomson, 1998) was applied to the combined normalized (by standard deviation) monthly time series of physical (temperature and salinity)

and chemical (oxygen, inorganic nutrients) data sets. It was performed at all depths in order to summarize the main signal of physical chemical co-variability in the water column. The first and second EOF analysis was applied for two periods (1996-2011 and 2000-2011) of time in order to evaluate the impact of the 1997-1998 El Niño on the statistics and grasp changes in the relationship between variables. The Pearson correlation coefficient (r) was calculated between series.

**2.4 Intraseasonal Equatorial Kelvin Waves (IEKW) and El Niño indices**

The amplitude of IEKW is derived from an Ocean General Circulation Simulation named Mercator. This simulation has been validated from observations in Mosquera-Vásquez et al. (2014), which indicates that it has comparable skill than the SODA oceanic reanalysis in the near equatorial region (Carton and Giese, 2008).

The method for deriving the wave coefficient consists in projecting the pressure and current anomalies from the model between 15°S and 15°N onto the theoretical vertical mode functions obtained from the vertical mode decomposition of the mean stratification. Kelvin wave amplitude is then obtained by projecting the results onto the horizontal modes at each grid point in longitude. The method has been shown to be successful in separating first and second baroclinic waves (Dewitte et al., 1999, 2008) that propagate at different phase speeds and impact the Peru coast in a very specific way (Illig et al., 2014). In particular, due to the sloping thermocline from west to east along the equator, the second baroclinic mode Kelvin wave is more energetic and influential on the upwelling variability off the Peruvian coast (Dewitte et al., 2011, 2012). For the correlation analysis with the dissolved oxygen data, we select the

IEKW amplitude (in cm) at 90°W for the first and second baroclinic modes (hereafter IEKW_1

and IEKW_2)

In order to diagnose the large scale interannual variability in the tropical Pacific, we also use the Oceanic El Niño Index (ONI) provided by the national Weather Service NOAA

(NOAA, CPC. 2015); and two other indices recently proposed by Takahashi et al. (2011) that characterise the variability associated to extreme EL Niño variability (E index) and the variability associated to Central Pacific and La Niña events (C index). Conveniently, these two indices are independent (uncorrelated) and derived from the EOF analysis of the SST anomaly in the tropical Pacific (see Takahashi et al. (2011) for details). The HadISST data set is used (Rayner et al., 2003) to derive them over the period 1950-2015, but only the values of the period

1996-2011 are used here.

**3. Results**

**3.1. Oceanographic dynamics off central Peru (Callao, 1996-2011)**

Vertical distributions of temperature (a) and salinity (b) off Callao (St 4) during 1996-

2011 years are shown in Figure 2. The data set collected in this coastal and shallow station shows a water column with strong interannual signal, with significant changes in the 15º C

isotherm depth. Here, the 15º C isotherm depth is considered as a proxy of the lower limit of the thermocline position, from 20 m to more than 100 m like during the 1997-1998 El Niño.

Under "normal" conditions, cold (< 15º C) and relatively salty (34.8-35.1) subsurface waters, characteristics of the Equatorial Subsurface Waters (ESSW), were dominant. Maximum temperatures at surface (up to 22º C, Figure 2a) occurred during summer and spring periods, when a shallow mixed layer occurs (up to 20 m water depth 15°C, Figure 3a). The climatology of the 15° C isotherm depth (Z_15°C) indicates a semi-annual component (Figure 3b), shallower in early fall and spring and deeper in winter during more intense upwelling events.

The time series of temperature (Figure 2a) shows a strong deepening in the 15º C

isotherm (see also Figure 3a) from the end of 1997 until the beginning of 1998. This was a consequence of warm and salty (>35.2) Subtropical Surface Waters (SSW) penetration into the coast through transport and the crossing of the downwelling Kelvin wave and associated extra- tropical Rossby wave. This situation corresponds to the impact of the intense 1997-1998 El

Niño, that switched off the coastal upwelling characteristics during almost one year. Note that the disappearance of the 15°C isotherm in the first 100 m took place in early 1997 (around

April), well ahead the El Niño peak phase (around November).

Between 1999 and 2001 the 15° C isotherm depth presented lower interannual variability and in general a shallow position. A slight deepening of the thermocline takes place at the beginning of 2002, 2005, during winter of 2006, 2008 and 2009. This thermocline deepening was coincident with high salinity values and the occurrence of weak or moderate El

Niño conditions.

**3.2 The dissolved oxygen and the Oxygen Minimum Zone (OMZ) variability**

The dissolved oxygen distribution shows a similar pattern to the thermohaline variables with strong anomalies during the 1997-1998 El Niño and the subsequent 1999-2001 La Niña event, (Figure 2c and 3). Shallow positions (20-40 m) of the oxycline (iso-oxygen of 45 μmol

$kg^{-1}$) and OMZ upper boundary (22.5 μmol $kg^{-1}$ iso-oxygen) were registered under active upwelling of ESSW and during the 1999-2001 La Niña event. On average, this condition is characteristic not only during summer-fall periods (see the climatological annual pattern, Figure

3b and c) but also in springtime suggesting a semi-annual component. During these seasons oxygen poor waters can intercept the euphotic layer and also the continental shelf, promoting suboxic and even anoxic conditions in the underlying surface sediments ($O_2$ < 8.9 μmol $kg^{-1}$,

Fig. 3a). During austral winter (July-August) the oxygen deficient waters deepen (up to 70 m depth, figure 2c and 3).

At interannual scale, during the strong 1997-98 El Niño, a significant deepening of the oxycline and OMZ upper boundary is observed, when well-oxygenated Surface Subtropical

Waters (SSW) occupied the water column down to at least 100 m depth. Besides the strong amplitude of the oxygen fluctuation associated to the 1997-98 El Niño, an interesting feature from the OMZ temporal distribution is the relatively large intraseasonal (i.e. periods lower than

1 year) variability since 2000 (see 2006, 2008, 2009, 2011 Figure 2 and 3). The relationship between $O_2$ anomalies and thermocline during the 1997 - 98 El Niño (i.e. positive $O_2$ anomalies associated to a deepening of the thermocline) breaks down afterwards for some events. Before

2000, the oxycline and the OMZ's upper boundary present a significant correlation with the

15ºC isotherm (r = 0.43, v-p < 0.01 and r = 0.67, v-p < 0.01 respectively), after the 2000, the correlation drops down (r = 0.12, v-p > 0.01(N.S) and r = 0.44, v-p < 0.01, respectively)

suggesting a decoupling (in the linear sense) between $O_2$ and thermocline. The position of the

OMZ upper limit show a negative trend estimated by a linear fit in -0.02 m/decade between

1996 and 2011, and in -0.12 m/decade after 2000 that suggest a long-term deepening of the oxygen deficient waters.

**3.3 Nutrients and biogeochemical activity**

The time series of inorganic nutrients vertical distribution off Callao are shown in Figure

4. Nitrate and nitrite concentrations ranged from ca. 0.0 to 27.0 µmol L$^{-1}$ and ca. 0.2 to 9.0 µmol

L$^{-1}$ values respectively. Lower nitrate values are present at surface and bottom waters, particularly during summer and fall periods, while maximum nitrite values appear at subsurface waters in opposite relationship with nitrate levels. During winter periods maximum nitrate concentrations characterize the entire water column (> 15 µmol L$^{-1}$), coincident with the period of maximum upwelling intensities. The vertical distributions of silicate and phosphate exhibit a similar pattern than nitrate.

Nutrient time series also present a strong interannual signal mostly prominent during the 1997 – 98 El Niño event with low nitrate concentrations (< 10 µmol L$^{-1}$) coincident with minimum and even zero nitrite values and low silicate and phosphate levels (<10 µmol L$^{-1}$ and

1 µmol L$^{-1}$ respectively; Figure 4a, b). Between 1999 and 2001 nitrate concentrations were also lower than 10 µmol L$^{-1}$ on average, but in contrast with the previous El Niño episode, subsurface nitrite reached maximum values (up to 9 µmol L$^{-1}$) coincident with an intense OMZ

development and shallow thermocline. Silicate depletion is observed near the surface, while phosphate appears as a non-limiting nutrient in the surface waters. Elevated phosphate concentration in the surface waters is typical near the coast, whereas nitrate is depleted at surface and in the water column (Kock et al., 2016).  At subsurface, high silicate (> 25 µmol L$^{-}$

$^{1}$) and phosphate (3 µmol L$^{-1}$) concentrations were observed. After 2000, nitrate concentrations registered the highest nitrate values (> 20 µmol L$^{-1}$) for the entire time series. The variability of nitrite concentrations also increased after 2000; high nitrate levels were coincident with lower nitrite and silicate concentrations (Figure 4).

In order to explore some biogeochemical activity related with the nitrogen cycle and the

OMZ variability off Callao, the Ndef in the water column is estimated (Figure 5). Ndef values range from negative (-5 µmol L$^{-1}$), indicative of low nitrate consumption, up to 40 µmol L$^{-1}$

corresponding to conditions of high deficiency in nitrate. The Ndef, particularly at subsurface, exhibits a clear interannual signal with minimum values (zero-negative) during the 1997-1998

El Niño coincident with well-oxygenated waters (Figure 3). In contrast, between 1999 and 2001

years, maximum Ndef (30-40 µmol L$^{-1}$) occurred under a shallow and well-developed OMZ.

Nitrate reduction during this period was associated to high nitrite at subsurface (Figure 4).

After 2000, the Ndef water column conditions were highly variable coincident with the variability in the OMZ distribution. Strong deficient conditions were registered in 2005, 2007

and at the end of 2011 coincident with the strong La Niña conditions. Ndef at subsurface (50

and 90 m depth) was significantly correlated with the 15ºC isotherm (r= 0.43, v-p < 0.01) and with the OMZ though the correlation is relatively low (r 0.28, v-p < 0.01).

**3.4 Equatorial forcing of the OMZ**

In this section, we first document the evolution of the IEKW activity during 1996-2011

and then interpret the variability of the biogeochemical parameters off Peru documented above in the light of the characteristics of the remote equatorial forcing. The evolution of the amplitude of the first and second baroclinic modes Kelvin waves (IEKW_1 and IEKW_2) at 90°W in terms of sea level anomalies is shown in Figures 6 a,b. These waves differ from their vertical structure and phase speed, and are the most energetic along the equator. They transmit their energy along the coast in the form of coastal trapped Kelvin wave and can trigger extra-tropical

Rossby waves (Clarke and Shi, 1991). It is assumed that waves with amplitude larger than one standard deviation over the study period (see horizontal lines in figure 6a,b) are downwelling

Kelvin waves, whereas amplitudes more negative than -1 standard deviation correspond to upwelling Kelvin waves. Therefore, coastal-trapped downwelling (upwelling) waves will tend to reduce (increase) wind-driven coastal upwelling near Callao.

Our data reveal that the IEKW_2 is delayed by 1 month compared to the IEKW_1, consistent with the difference in phase speed of the waves and their propagation from the central equatorial Pacific up to 90°W. Maximum correlation between both time series was before 2000

(r 0.67, v-p< 0.01), being significantly lower (r 0.42, v-p< 0.01) for the period after 2000. The lower coherency between both Kelvin wave modes can be interpreted as resulting from non- linear interactions of the waves with the mean thermocline near 120°W (see Mosquera-Vásquez et al., (2014)). While the variability of both modes at 90°W looks similar, owned to their different vertical structure, their impact of the regional oceanic circulation off Peru is distinct, with in particular the second baroclinic mode Kelvin wave being trapped at a latitude closer to the equator than the first baroclinic mode Kelvin waves (Clarke and Shi, 1991). The second baroclinic mode Kelvin wave is also associated to lower frequency variability than the first baroclinic Kelvin wave as revealed by the global wavelet spectra analyses of the IEKW_1 and

IEKW_2 timeseries at 90°W (Figure 6 a,b) consistently with Dewitte et al. (2008). The equatorial Kelvin wave experiences modal dispersion near 120°W (see Mosquera-Vásquez et al 2014) so that the amplitude and coherence of IEKW_2 and IEKW_1 can change along their propagation to the east of 120°W. This explains the different spectrum of variability of IEKW_2

and IEKW_1 although both waves exhibit an energetic peak around 50 days that corresponds to the forcing by intraseasonal atmospheric fluctuations in the western- central Pacific (e.g.

Westerly Wind Bursts). It is noteworthy that the IEKW_2 is negatively skewed from 2000

(normalized skewness = -0.8910 cm) and there is a negative trend of the upwelling events from

2000 (trend = -0.0177 cm/decade), features that are also encountered for the Z_15°C (Fig. 6c)

(normalized skewness = -1.330 m and trend = 0.0250 m/decade). Positive anomalies of IEKW

(Figure 6 a, b) are associated to a deepening of Z_15°C (<0), as was observed during the 1997-

1998 El Niño, the weak 2002-03 El Niño and during 2006, 2008 and 2010 warm seasons.

During these periods IEKW_1 and IEKW_2 are in phase with comparable amplitude and the

Z_15°C and the Z_ZMO (Fig. 6 d) are out of phase. The IEKW_1 and IEKW_2 are highly correlated with Z_15°C and Z_OMZ variables, but we find that IEKW_2 has a stronger relationship with the Z_15°C and Z_OMZ (r -0.54, -0.40 respectively, v-p<0.01) than IEKW_1

(r -0.34, -0.23 respectively, v-p<0.01).

The global wavelet spectrum analysis of Z_15°C and Z_OMZ (Fig. 6 h, i) reveals a rich spectrum of variability that encompasses low frequency and intraseasonal frequencies. The intraseasonal frequencies (~50-180 days) are consistent with the forcing by IEKW_2 while the low-frequency timescales result from the combined effect of both waves. Consistently the nitrogen deficit at 50 m (N_def; Fig. 6d) exhibits a comparable spectrum than Z_15°C and

Z_OMZ though with a significant dominant peak at ~4 years.

In order to further document the variability in physical and biogeochemical variables and synthesize their relationship, an EOF analysis is performed combining all normalized time series. The results are presented in Figure 7 and table 1. The first EOF (EOF_1) mode of the combined temperature, salinity, oxygen, nitrate and nitrite explains 48 % of the total variability and covers the large fluctuations associated to the 1997- 1998 El Niño event (Figure 7 b).

Fluctuations after 2000 are more in phase with the C index, with a relatively low correlation (r

-0.49; Table 2) suggesting the influence of Central Pacific and La Niña events on the OMZ

dynamics. Interestingly the amplitude of the mode gives a similar weight (in absolute value) to temperature (-0.58), salinity (-0.46), oxygen (-0.48) and nitrite (0.46) (see first column of Table

1).

The second EOF mode (EOF_2) accounts for 24% of the explained variance and exhibits higher-frequency timescales than EOF_1 with no clear relationship with the 1997-

1998 El Niño event since its correlation with the E index (that accounts for extreme El Niño events) is not significant (Figure 7b, Table 2). The relative amplitude of the variables for this mode also indicates a larger contribution of biogeochemical variables. The amplitude of nitrate (-0.86) and oxygen (0.27) were larger than the physical variable amplitudes (temperature, 0.17

and salinity 0.09). Interestingly EOF_2 is not linearly related to the Kelvin wave modes and the

El Niño indices (Table 2) over the entire period. Only the correlation with the C index is significant but remains relatively low (-0.35).

The large signature of the strong 1997 - 98 El Niño event onto the EOF modes questions to which extent the statistics is impacted by the consideration of this event. In order to test the sensitivity of the results to the period under consideration and attempt to isolate features independent of the strong 1997/98 El Niño event, the similar combined EOF analysis was performed over the period 2000-2011 (Figure 7c). The results indicate that the EOF modes capture significantly distinct characteristics than the EOF modes obtained for the full period.

The most visible change is that the modes tend to capture higher frequencies, which is also revealed through spectral analysis (not shown). The amplitude of the mode for each variable is also drastically changed compared to the analysis over the full period with in particular, for

EOF_2,the amplitude for temperature (0.43) being increased along with the one of oxygen (0.61) (see Table 1). It suggests a distinct statistical relationship between variables over the period after 2000. Another noticeable change is in the relationship of the P and C time series El

Niño indices and Kelvin wave coefficients. In particular, the EOF_2 is also correlated to

IEKW_2, which was not the case for the analysis over the full period. Overall, the results of the

EOF analysis suggest two different regimes of OMZ dynamics related to distinct physical equatorial forcing.

**4. Discussion**

The time series of oxygen and nutrients between 1996 and 2011 in the central area of

Peru reveal a rich spectrum of variability in the position and intensity of the OMZ with timescales spanning the intraseasonal, seasonal (with in particular a marked semi-annual cycle)

to interannual frequencies. Consistently with a previous study (Gutiérrez et al., 2008), such variability can be largely interpreted as resulting from the remote equatorial forcing owned to the efficient oceanic teleconnection off Peru.

Firstly, on a **seasonal** scale, the oxygen data exhibit a marked semi-annual pattern, similar to the seasonality of the 15C isotherm depth. The signature in the isotherm is consistent with the equatorial oceanic teleconnection considering that thermocline variability along the equator has a semi-annual cycle (Yu and McPhaden, 1999, Ramos et al., 2006). A shoaling of the oxycline and OMZ was observed during summer/fall (January-April) and early spring (October), up to < 50m, usually overlapped with the periods of the highest levels of chlorophyll-

*a* and primary productivity rates in the area (Pennington et al., 2006, Echevin et al., 2008). Low nutrient values at sea surface are consistent with phytoplankton uptake, while low nitrate can dominate at subsurface because of the high nitrate reduction activity and nitrogen loss under more intense OMZ conditions (Graco et al., 2007, Echevin et al., 2008; 2014). During austral winter (July-August), under strong upwelling favorable winds, the opposite was observed with a deepening of the oxycline and the OMZ (> 40-50 m) and the increase of nutrients. This result is consistent with models that show a dual role of winds in winter: stronger winds favor upwelling and nutrient availability while increased high-frequency wind activity produced mixing and oxygenation in the water column (see Vergara et al., 2016). In addition, a decrease in primary productivity in this season determine a lower nutrient uptake and lower availability of organic matter and in consequence less oxygen consumption (Graco et al., 2007; Echevin et al., 2008).

Superimposed to the seasonal/semiannual variability stronger changes in the OMZ

conditions and nutrients occur at interannual scales. The study period appears to cover two contrasting biogeochemical regimes. (1) one associated to the strong 1997 – 1998 El Niño and the subsequent La Niña events representing a marked ENSO cycle and (2) the period after 2000

characterized by episodic weak Warm and strong Cold Events with strong intraseasonal variability in the environmental forcing (i.e. equatorial Kelvin wave).

Under *the strong El Niño* **1997-1998 period**, the chemical time series (oxygen and nutrients) evidence the modulation of the biogeochemistry activity by remote forcing from the

Equatorial Pacific. The water column is dominated by warm, oxygenated and nutrient- poor water masses, which resulted from the onshore intrusion of Subtropical Surface Waters (SSW)

in the central area off Peru during El Niño events (Morón, 2000, Morón and Escudero, 1991); and it is associated to the geostrophic adjustment of the circulation in the form of downwelling coastal Kelvin and extra-tropical Rossby waves (Dewitte et al., 2012). It is shown that the

Kelvin waves of both modes are in phase and correspond to marked downwelling conditions of comparable magnitude. The disappearance of the 15°C isotherm in the first 100 m took place in the early 1997 (around April) well ahead the El Niño peak phase (around November)

associated to the impact of the downwelling Kelvin waves that were triggered in December and

March 1997 in the western Pacific (Dewitte et al., 2003). These waves deepened the coastal thermocline, initiating the anomalous conditions prior to the development of El Niño (Ramos et al., 2008). The biogeochemical activity was clearly coupled to the physical forcing during this strong event. There was a significant deepening of the nutricline (> 80 m depth) associated to a thermocline deepening and the disappearance of oxygen deficient waters (< 45 µM). Our

EOF analysis and the high correlation with the El Niño indices (ONI and E, Table 1) point to this interannual modulation. Similar conditions were reported for the 1982-1983 strong El Niño when a significant deepening of the thermocline, oxycline and nutricline, and a general increase in oxygen concentration in the subsurface layers were observed off Peru (Guillén et al 1989).

These conditions appear unfavorable for the development of an important primary productivity, due to a lower nutrient availability in the surface layer. Low values of Ndef associated to low nitrate and almost zero nitrite concentrations suggest lower denitrification and/or anammox activity. The significant effect on denitrification in the eastern South Pacific Ocean due to changes in the equatorial winds during El Niño was previously described by Codispoti et al.

(1988) as a large-scale response to EN. On the continental shelf off Callao, Graco et al (2008), showed the occurrence of interannual variability in denitrification and anammox rates, with a significant decrease in nitrogen loss processes under El Niño coupled also with a decrease in primary productivity and an increase in oxygen levels. Conversely, during the **cold phase of**

**the ENSO cycle, 1999-2001 La Niña event**, there is a shoaling of the OMZ upper boundary and the nutricline (silicates and phosphate) coupled with a shallow thermocline and associated with IEKW_1 and 2 activities, in phase but dominated by negative anomalies. Codispoti et al., (1988) proposed a thermocline "overshoot" in the years following an El Niño event, and Guillén and Calienes (1981) suggest that cold anomalies can occur up to three years after an El Niño, as observed in this study. In fact, during 1999-2001 our results show a dominance of cold, oxygen deficient and nutrients rich upwelling waters off the central Peru. The co-occurrence of a shallow and intense OMZ (< 20 m depth) with low nitrate values (< 10 µmol L$^{-1}$), subsurface nitrite maximum (up to 9 µmol L$^{-1}$) and high Ndef point to an important biogeochemical activity during the cold periods or la Niña event not previously reported. These conditions suggest a high organic matter remineralization coupled to an intense oxygen demand and denitrification/anammox processes in the area. High nitrite pools in the water column were described as a typical feature under oxygen deficient waters (Deuser et al., 1978; Naqvi, 1991)

and a tracer of denitrification (Codispoti and Packard, 1980; Codispoti et al., 1985; 1986; 2001)

and anammox activity (Hammersley et al., 2007; Lam et al., 2009; Lam and Kuypers, 2011) off

Peru. Similar results were observed in other upwelling ecosystems (Calvert and Price, 1971,

Morales et al., 1996; Naqvi et al., 1994) and in the Eastern tropical South Pacific (Codispoti and Christensen, 1985; Tyrrell and Lucas, 2002).

**After 2000,** the oceanographic conditions responded to ***"episodic weak Warm and***

***strong Cold Events"*** and were characterized by an apparent decoupling between the physics (thermocline) and the OMZ, suggesting a non-linear relationship between the OMZ dynamics and the remote equatorial forcing over this period. Our results suggest that the biogeochemical variables appear modulated by an intense seasonal variability to which is superimposed higher frequency variability associated to upwelling (downwelling) IEKW activity coincident with a the higher frequency of occurrence of Central Pacific El Niño events for the last decade (2002-

2004, 2006, 2008/2009; Yu and Kim, 2013). A less intense OMZ (oxygen > 10 µmol L$^{-1}$), higher nitrate levels (> 20 µmol L$^{-1}$), almost zero nitrite values and low Ndef at the subsurface waters appear under these intermittent "warm periods" characterized by downwelling IEKW

activity, like in 2002, 2006, and 2008. The low Ndef and high nitrate concentrations suggest a low nitrogen loss in the area under these conditions. On the contrary, episodic "cold events", as in 2005, 2007 and 2010-2011, are associated with negative anomalies of the IEKW (upwelling), a well-developed OMZ and an intense nitrogen recycling as suggested by the high Ndef values.

While the length of the data set is a limitation (only 12 years of data after 2000), the analysis suggests a long-term trend in oxygen concentration near the oxycline (decreasing trend) that is consistent with an increasing trend in the amplitude of the upwelling events associated to the second baroclinic mode Kelvin wave (IEKW_2). In particular, after 2000, the IEKW_2 is negatively skewed (-0.8910 cm) and upwelling events show a negative tendency (-0.0177

cm/decade). Note that the IEKW_2 is more likely to be influential on the coastal circulation than the first baroclinic mode Kelvin wave due to its vertical structure having two nodes on the vertical, which is similar to the dominant empirical modes along the coast in this region (see

Echevin et al. (2014)). Similar desoxigenation trends in the last decades have been recently reported for other coastal upwelling systems (California Current, *Bograd et al.,* 2008; Benguela

Current, Monteiro pers. com.) and they could trigger changes in phytoplankton size and/or community structure, as was observed in California Current (Chavez pers.com.).

We now discuss limitations of our analysis. While our results suggest that the OMZ

variability off Callao can be understood in terms of the oceanic equatorial teleconnection, our interpretation of the OMZ variability does not consider aspects of the wind forcing, although the latter is highly variable in the central Peru region and is influential on the upwelling dynamics. While during El Niño events, there is in general a weakening of the upwelling favorable winds at regional scale owned to the relaxation of the South Eastern branch of the trade winds, near the coast winds can intensified locally owned to the effect of the underlying warm waters (Dewitte and Takahashi, 2016, submitted). To which extent such anomalous winds influence the local oceanic circulation and associated biogeochemical response remains to be investigated. Considering the limited knowledge on this aspect, we have not introduced the analysis of the local wind forcing at interannual timescale here. Regarding intraseasonal wind variability, Dewitte et al. (2011) reports two regimes of along-shore wind variability off Callao.

One regime associated to extra-tropical storms activity modulating the South Pacific

Anticyclone and corresponding to frequencies ranging to 1 to 25 days, and a second regime with frequencies ranging from 30 to 90 days that is associated to atmospheric teleconnections from the western tropical Pacific. Illig et al. (2014) evaluate the influence of these two wind regimes on the oceanic teleconnection at intraseasonal timescales based on the experimentation with a regional oceanic model and found that they are weakly influential on the propagation of the Kelvin wave of equatorial origin. Therefore, the limitation of not including the winds in our analysis may not be detrimental to our results. Note also that Gutiérrez et al. (2008), estimated that 43% of the temporal variability in the oxygen regime over the continental shelf off Callao is explain by the remote forcing. Echevin et al. (2014) also show evidence that subsurface nutrient and chlorophyll intraseasonal variability are mainly forced by the coastally trapped waves triggered by intraseasonal equatorial Kelvin waves reaching the South-American coast.

In the central and southern part of Peru, that include our study area (Callao 12° 02´S), IEKW- forced CTW signature on the circulation and aspects of the biogeochemistry thus emerges as dominant process over the local wind impact at the timescales of interest in this paper (i.e.

periods > 1 month).

In addition, while the global trend in the open ocean appear to be an expansion of the oxygen, deficient waters, particularly in the tropical oceans during the past 50 years (Stramma et al., 2008; 2010), our results suggest a long-term deepening of the oxygen deficient waters from 2000 (-0.12 m/decades). In fact, in the coastal areas many questions remain open related with short-term temporal variability and the onshore-offshore physical and biogeochemical coupling dynamics that regulate in a complex interplay the intensity and distribution of the OMZ and can determine different biogeochemical scenarios with a potential impact for the coastal human communities (Gutiérrez et al., 2011b). The data presented in this paper could therefore serve as a useful benchmark for testing paradigms for explaining OMZ variability under different mean state conditions, changes in nutrients condition that can modulate the phytoplankton response and productivity, as well as for the validation of regional coupled modeling platforms intended to address low-frequency variability of the OMZ in the Eastern Pacific.

To conclude, the study period appears to illustrate distinct regimes of the OMZ dynamics and biogeochemical activity off Peru. One regime with a strong asymmetry associated to the extreme Eastern Pacific 1997- 1998 El Niño when the OMZ disappeared in the upper layer and the subsequent intensification of the OMZ and nitrogen lost during 1999-2001 La Niña. Other regime since 2000 characterized by a strong intraseasonal variability in the intensity of the OMZ and the availability of nutrients, loss nitrogen processes; and a tendency for a decoupling between the chemistry and the physical forcing associated with weaker but more frequent warm events and a higher frequency variability of upwelling (downwelling) IEKW activity.

*Acknowledgements*

This research was financed and conducted in the frame of the Instituto del Mar del Peru

(IMARPE). We thank Carlos Robles and Miguel Sarmiento, the technical chemical staff.

Thanks the crew of the IMARPE VIII and the SNP-2 research vessels and all the scientific

colleagues that help us. The first author thanks the financial support of EU-project CENSOR

(Climate variability and El Nino Southern Oscillation: Impact for natural resources and

management, contract 511071), the WP3 of the LMI DISCOH project (cooperation agreement

between IMARPE and IRD). We grateful to the two anonymous reviewers for their constructive

comments that helped improving the manuscript. This work is a contribution of the project

"Integrated Study of the Upwelling system off Peru" developed in the Direction of

Oceanography and Climate Change Research.

**Table 1**:
Percentage of explained variance of the EOF modes (first line) and eigenvectors of the EOF
modes (i.e. amplitude of the EOF mode in terms of the different variables).

|  | EOF1 | EOF2 | EOF1_2000-2011 | EOF2_2000-2011 |
|---|---|---|---|---|
| Percentage of explained variance | 48 % | 24% | 45% | 28% |
| Temperature | -0.58 | 0.17 | -0.33 | 0.43 |
| Salinity | -0.46 | 0.09 | -0.36 | 0.11 |
| Oxygen | -0.48 | 0.27 | -0.37 | 0.61 |
| Nitrate | -0.01 | -0.86 | -0.59 | -0.64 |
| Nitrite | 0.46 | 0.39 | 0.51 | 0.06 |

**Table 2** : Coefficient correlation Pearson at 95% confidence interval between the first two PC
timeseries and the Kelvin wave and El Niño indices.

|  | EOF1 | EOF2 | EOF1_2000/2011 | EOF2_2000-2011 |
|---|---|---|---|---|
| IEKW_1 | -0.34 | N.S. | N.S . | N.S. |
| IEKW_2 | -0.60 | N.S. | -0.37 | 0.40 |
| ONI | -0.65 | N.S. | -0.57 | N.S. |
| E | -0.73 | N.S. | N.S. | 0.37 |
| C | -0.28 | -0.35 | -0.49 | N.S . |

**Figure Legends**

**Figure 1.** Location of the sampling station (St 4; 20 nm, 145 m depth) in the coastal upwelling ecosystem off central Peru, Callao (12º 02´ S).

**Figure 2.** Time series of temperature (°C) (a), salinity (b) and dissolved oxygen ($\mu$mol kg$^{-1}$) (c)

during the 1996-2011 study years.

**Figure 3.** Time series of the depth of the 15º C isotherm, oxycline (45.0 $\mu$mol kg$^{-1}$) and the

OMZ upper boundary (22.5 $\mu$mol kg$^{-1}$) (a) and the seasonal pattern of the 15° C isotherm depth (b), the oxycline depth (45.0 $\mu$mol kg$^{-1}$) (c) and the upper boundary of the OMZ depth (22.5

$\mu$mol kg$^{-1}$) (d) during 1996-2011 time series.

**Figure 4.** Time series of nitrate (a), nitrite (b), silicate (c) and phosphate (d) during 1996-2011.

Contours in solid and dashed lines indicate the 22.5 $\mu$mol kg$^{-1}$ and 45 $\mu$mol kg$^{-1}$ oxy-lines, respectively. Units are $\mu$M L$^{-1}$.

**Figure 5.** Time series of N deficit ($\mu$mol L$^{-1}$) at St. 4 off Callao during 1996-2011.

**Figure 6.** Evolution of the (a) Intraseasonal Equatorial Kelvin Waves (IEKW) anomalies at

90°W for the first (IEKW_1) and second (IEKW_2) baroclinic modes. Units are cm (equivalent sea level). The standard deviation is indicated by the horizontal dashed lines. Time resolution of the data is every 5 days, (b) depth of the thermocline; (c) OMZ's upper boundary depth and (d) Fixed Nitrogen deficit (Ndef) at 50 m (e ) at St. 4 off Callao during 1996-2011. On the right hand side of each timeseries, the global spectrum wavelet analysis are shown (f-j).

**Figure 7.** Temporal series of (a) the C and E indices (b) the principal component of the first two dominant EOF modes of the temperature, salinity, oxygen, nutrient and nitrite combined timeseries over the period 1996-2011(c) same as (b) but for the period 2000-2011. Timeseries were normalized by their standard deviation prior to analysis.

[Figure]

**Figure 1.**

[Figure]

[Figure]

**Figure 2.**

[Figure]

**Figure 3.**

[Figure]

**Figure 4.**

[Figure]

**Figure 5.**

[Figure]

**Figure 6.**

[Figure]

**Figure 7.**

---

## Author Response (AR3)

Lima, the 31 August, 2017

Michelle Graco
Department of Oceanography
Instituto del Mar del Perú (Imarpe)
Esq. Gamarra y Gral Valle s/n
Chucuito-Callao, Peru
Phone/Fax: 0051-1-429 6069
e-mail: mgraco@imarpe.gob.pe

Dear Dr Herndl,

We highly appreciated all the comments and suggestions from all the reviewers during the process about our manuscript entitled "THE OMZ AND NUTRIENT FEATURES AS A SIGNATURE OF INTER-ANNUAL AND LOW-FREQUENCY VARIABILITY OFF THE PERUVIAN UPWELLING SYSTEM" by M. Graco and co-authors.

We have revised and corrected the manuscript following all the reviewer comments and technical corrections.

We are grateful to the editorial office and the dedicated work of each reviewer that improved our article. Thank you very much.

Best regards

Michelle I. Graco

Michelle I. Graco

Remarks for technical corrections and proposed modifications:

-Abstract line 16 an instead of e. *It was corrected.*

-Line 19: write (12°02´S, 77° 29´W) instead (12°02´S, 20 mn). *It was modified.*

-Line 45 Tsuchiya instead of Tsuchida. *It was corrected.*

-Line 72. Moron, not Moron O. *It was corrected.*

-Line 167 according to references Mosquera Vasquez. *It was corrected.*

-Lines 446-447 Ref is not in the text. *It was removed.*

-Lines 475-476 move year to the end of the reference. *It was modified.*

-Lines 481-482 Ref is not in the text. *It was removed.*

-Fig. 1. Coordinates of the inset *modified to be readable.*

-Fig 2-5. The size of numbers in the figures were *modified.*